# Dopaminergic challenge dissociates learning from primary versus secondary sources of information

Alicia J Rybicki*, Sophie L Sowden, Bianca Schuster, Jennifer L Cook

Centre for Human Brain Health, School of Psychology, University of Birmingham, Birmingham, United Kingdom

**Abstract** Some theories of human cultural evolution posit that humans have social-specific learning mechanisms that are adaptive specialisations moulded by natural selection to cope with the pressures of group living. However, the existence of neurochemical pathways that are specialised for learning from social information and individual experience is widely debated. Cognitive neuroscientific studies present mixed evidence for social-specific learning mechanisms: some studies find dissociable neural correlates for social and individual learning, whereas others find the same brain areas and, dopamine-mediated, computations involved in both. Here, we demonstrate that, like individual learning, social learning is modulated by the dopamine D2 receptor antagonist haloperidol when social information is the primary learning source, but not when it comprises a secondary, additional element. Two groups (total N = 43) completed a decision-making task which required primary learning, from own experience, and secondary learning from an additional source. For one group, the primary source was social, and secondary was individual; for the other group this was reversed. Haloperidol affected primary learning irrespective of social/individual nature, with no effect on learning from the secondary source. Thus, we illustrate that dopaminergic mechanisms underpinning learning can be dissociated along a primary-secondary but not a social-individual axis. These results resolve conflict in the literature and support an expanding field showing that, rather than being specialised for particular inputs, neurochemical pathways in the human brain can process both social and non-social cues and arbitrate between the two depending upon which cue is primarily relevant for the task at hand.

*For correspondence:
axr783@bham.ac.uk

**Competing interest:** The authors declare that no competing interests exist.

## Editor's evaluation

This work has important implications on how we view and understand social and individual learning with respect to dopamine processing in the human brain. This study, supported by a well-controlled experimental design, clear hypothesis testing, and rigorous model-based analyses, revealed that the dopamine system is involved in learning from a primary source as opposed to a secondary source, irrespective of social or non-social individual learning. This work encourages new investigations into testing when and how different neuromodulator systems may converge or diverge in guiding social versus non-social learning.

## Introduction

The complexity and sophistication of human learning are increasingly appreciated. Enduring theoretical models illustrate that learners utilise 'prediction errors' to refine their predictions of future states (e.g., Rescorla–Wagner [RW] and temporal difference models; *O'Doherty et al., 2003*; *Rescorla and Wagner, 1972*; *Schultz et al., 1997*; *Sutton and Barto, 2018*). An explosion of studies, however,

illustrates that this simple mechanism lies at the heart of more complex and sophisticated systems that enable humans (and other species) to learn from, keep track of the utility of, and integrate information from, multiple learning sources (*Behrens et al., 2009*; *Biele et al., 2009*; *Li et al., 2011*), meaning that one can learn from many sources of information simultaneously (*Daw et al., 2006*). Such complexity enables individuals to, for example, rank colleagues according to the utility of their advice and learn primarily from the top-ranked individual (*Kendal et al., 2018*; *Laland, 2004*; *Morgan et al., 2012*; *Rendell et al., 2011*) whilst also tracking the evolving utility of advice from others (*Behrens et al., 2008*; *Biele et al., 2011*). Recent studies have further revealed that learning need not rely solely on directly experienced associations since one can also learn via inference (*Bromberg-Martin et al., 2010*; *Dolan and Dayan, 2013*; *Jones et al., 2012*; *Langdon et al., 2018*; *Moran et al., 2021*; *Sadacca et al., 2016*; *Sharpe and Schoenbaum, 2018*). This growing appreciation of the complexity and sophistication of human learning may help to explain contradictory findings in various fields. Here, we focus on the field of social learning.

The existence in the human brain of neural and/or neurochemical pathways that are specialised for learning from social information and individual experience respectively is the topic of much debate (*Heyes, 2012*; *Heyes and Pearce, 2015*). Indeed, the claim that humans have *social-specific learning mechanisms* that are adaptive specialisations moulded by natural selection to cope with the pressures of group living lies at the heart of some theories of cultural evolution (*Kendal et al., 2018*; *Morgan et al., 2012*; *Templeton et al., 1999*). Since cultural evolution is argued to be specific to humans (*Richerson and Boyd, 2005*), establishing whether humans do indeed possess social-specific learning mechanisms has attracted many scholars with its promise of elucidating the key ingredient that 'makes us human'.

Cognitive neuroscience offers tools that are ideally suited to investigating whether the mechanisms underpinning social learning (learning from others) do indeed differ from those that govern learning from one's individual experience (individual learning). Cognitive neuroscientific studies, however, present mixed evidence for *social-specific* learning mechanisms. Some studies find dissociable neural correlates for social and individual learning (*Apps et al., 2016*; *Behrens et al., 2008*; *Hill et al., 2016*; *Zhang and Gläscher, 2020*). For example, a study by *Behrens et al., 2008* reported that whilst individual learning was associated with activity in dopamine-rich regions such as the striatum that are classically associated with reinforcement learning, social learning was associated with activity in a dissociable network that instead included the anterior cingulate cortex gyrus (ACCg) and temporoparietal junction. Further supporting this dissociation, studies have revealed correlations between personality traits, such as social dominance (*Cook et al., 2014*) and dimensions of psychopathy (*Brazil et al., 2013*) and social, but not individual, learning, as well as atypical social, but not individual, prediction error-related signals in the ACCg in autistic individuals (*Balsters et al., 2017*). Together, these studies support the existence of *social-specific* learning mechanisms. In contrast, other studies have reported that the same computations, based on the calculation of prediction error, are involved in both social and individual learning (*Diaconescu et al., 2014*), and that social learning is associated with activity in dopamine-rich brain regions typically linked to individual learning (*Biele et al., 2009*; *Braams et al., 2014*; *Campbell-Meiklejohn et al., 2010*; *Delgado et al., 2005*; *Diaconescu et al., 2017*; *Klucharev et al., 2009*). *Diaconescu et al., 2017*, for example, observed that social learning-related prediction errors covaried with naturally occurring genetic variation that affected the function of the dopamine system. Further supporting this overlap between social and individual learning, behavioural studies have observed that social and individual learning are subject to the same contextual influences. For example, *Tarantola et al., 2017* observed that prior preferences bias social learning, just as they do individual learning. Such findings promote the view that 'domain-general' learning mechanisms underpin social learning: we learn from other people in the same way that we learn from any other stimulus in our environment (*Heyes, 2012*; *Heyes and Pearce, 2015*). That is, there are no *social-specific* learning mechanisms.

One potential resolution to this conflict in the literature hinges on (1) an appreciation of the complexity and sophistication of human learning systems and (2) a difference in study design between tasks that have, and have not, found evidence of *social-specific mechanisms*. In studies that have linked social learning with the dopamine-rich circuitry typically associated with individual learning (and which are therefore consistent with the domain-general view), participants have been encouraged to learn *primarily* from social information. Indeed, in many cases the social source has been the sole

information source (*Campbell-Meiklejohn et al., 2010*; *Diaconescu et al., 2017*; *Klucharev et al., 2009*). For example, in the paradigm employed by Diaconescu and colleagues (2014, 2017), participants were required to choose between a blue and green stimulus and were provided with social advice which was sometimes valid and sometimes misleading; on each trial, participants received information about the time-varying probability of reward associated with the blue and green stimuli, thus participants did not have to rely on their own individual experience of blue/green reward associations and could fully dedicate themselves to social learning. That is, participants did not learn from multiple sources (i.e., social information *and* individual experience); participants *only* engaged in social learning. In contrast, in studies where social learning has been associated with neural correlates outside of the dopamine-rich regions classically linked to individual learning (and which are therefore consistent with the domain-specific view), social information has typically comprised a secondary, additional source (*Behrens et al., 2008*; *Cook et al., 2014*). Typically, the non-social (individual) information is presented first to participants, represented in a highly salient form, and is directly related to the feedback information. The social information, in contrast, is presented second, is typically less salient in form, and is not directly related to the feedback information. For example, in the Behrens et al. study (2008) (and in our own work employing this paradigm; *Cook et al., 2014*; *Cook et al., 2019*), participants were required to choose between two, highly salient, blue and green boxes to accumulate points. The boxes were the first stimuli that participants saw on each trial. Outcome information came in the form of a blue or green indicator, thus *primarily* informing participants about whether they had made the correct choice on the current trial (i.e., if the outcome indicator was blue, then the blue box was correct). In addition, each trial also featured a thin red frame, which represented social information, surrounding one of the two boxes. The red frame was the second stimulus that participants saw on each trial and indirectly informed participants about the veracity of the frame: if the outcome was blue *and* the frame surrounded the blue box, then the frame was correct. In such paradigms, participants must learn from multiple sources of information with one source taking primary status over the other. Consequently, in studies that have successfully dissociated social and individual learning the two forms of learning differ both in terms of social nature (social or non-social) and rank (primary versus secondary status). Thus, it is unclear which of these two factors accounts for the dissociation.

This study tests whether social and individual learning share common neurochemical mechanisms when they are matched in terms of (primary versus secondary) status. Given its acclaimed role in learning (*Glimcher and Bayer, 2005*; *Schultz, 2007*), we focus specifically on the role of the neuromodulator dopamine. Drawing upon recent studies illustrating the complexity and sophistication of human learning (*Daw et al., 2005*; *Gläscher et al., 2011*; *Moran et al., 2021*), we hypothesise that pharmacological modulation of the human dopamine system will dissociate learning from two sources of information along a primary versus secondary, but not along a social versus individual axis. In other words, we hypothesise that social learning relies upon the dopamine-rich mechanisms that also underpin individual learning when social information is the primary source, but not when it comprises a secondary, additional element. Such a finding would offer a potential resolution to the aforementioned debate concerning the existence of *social-specific* learning mechanisms.

Preliminary support for our hypothesis comes from three lines of work. First, studies have convincingly argued for flexibility within learning systems. For example, in a study by *Daw et al., 2006*, participants tracked the utility of four uncorrelated bandits, with particular brain regions – such as the ventromedial prefrontal cortex – consistently representing the value of the top-ranked bandit, even though the identity of this bandit changed over time. Second, studies are increasingly illustrating the flexibility of social brain networks (*Ereira et al., 2020*; *Garvert et al., 2015*). The medial prefrontal cortex (mPFC), for example, is not – as was once thought – specialised for representing the self; if the concept of 'other' is primarily relevant for the task at hand, then the mPFC will prioritise representation of other over self (*Cook, 2014*; *Nicolle et al., 2012*). Finally, in a recent study (*Cook et al., 2019*), we provided preliminary evidence of a catecholaminergic (i.e., dopaminergic and noradrenergic) dissociation between learning from primary and secondary, but not social and individual, sources of information. In this work (*Cook et al., 2019*), we employed a between-groups design, wherein both groups completed a version of the social learning task adapted from *Behrens et al., 2008* described above. For one group, the secondary source was social in nature (social group). For the non-social group, the secondary source comprised a system of rigged roulette wheels and was

thus non-social in nature. We observed that, in comparison to placebo (PLA), the catecholaminergic transporter blocker methylphenidate only affected learning from the primary source, which, in this paradigm, always comprised participant's own individual experience. Methylphenidate did not affect learning from the secondary source, irrespective of its social or non-social nature. That is, we found positive evidence supporting a dissociation between primary and secondary learning but no evidence to support a distinction between learning from social and non-social sources. Nevertheless, since we did not observe an effect of methylphenidate on learning from the (social or non-social) secondary source of information, this study was unable to provide positive evidence of shared mechanisms for learning from social and non-social sources. If it is truly the case that domain-general (neurochemical) mechanisms underpin social learning, it should follow that pharmacological manipulations that affect individual learning when individual information is the primary source also affect social learning when social information is the primary source.

The current (pre-registered) experiment tested this hypothesis by orthogonalising social versus individual and primary versus secondary learning. We perturbed learning using the dopamine D2 receptor antagonist haloperidol (HAL), in a double-blind, counter-balanced, PLA-controlled design. To test whether pharmacological manipulation of dopamine dissociates learning along a primary-secondary and/or a social-individual axis, we developed a novel between-groups manipulation wherein one group of participants learned primarily from social information and could supplement this learning with their own individual experience, and a second group learned primarily from individual experience and could supplement this learning with socially learned information. To foreshadow our results, we demonstrate that HAL specifically affects learning from the primary (not secondary) source of information. Bayesian statistics confirmed that the effects of haloperidol were comparable between the groups, thus, HAL affected individual learning when individual information was the primary source and, to the same extent, social learning when social information was the primary source. Our data support an expanding field showing that, rather than being fixedly specialised for particular inputs, neurochemical pathways in the human brain can process both social and non-social cues and arbitrate between the two depending upon which cue is primarily relevant for the task at hand (*Cook, 2014*; *Garvert et al., 2015*; *Nicolle et al., 2012*).

## Results

Participants (n = 43; aged 19–38, mean [standard error] $\bar{x}(\sigma_{\bar{x}})$ = 25.950 [0.970]; 24 males, 19 females; see Materials and methods) completed an adapted version of the behavioural task originally developed by *Behrens et al., 2008*. Participants were randomly allocated to one of two groups. Participants in the *individual-primary group* (n = 21) completed the classic version of this task (*Figure 1A*; *Behrens et al., 2008*) in which they were required to make a choice between a blue and green box in order to win points. A red frame (the social information), which represented the most popular choice made by a group of four participants who had completed the task previously, surrounded either the blue or green box on each trial, and participants could use this to help guide their choice. The actual probability of reward associated with the blue and green boxes and the probability that the red frame surrounded the correct box varied according to uncorrelated pseudo-randomised schedules (*Appendix 2—figure 1*). For the individual-primary group, the individual information (blue and green stimuli) was primary, and the social information (red stimulus) was secondary on the basis that the blue/green stimuli appeared first on the screen, were highly salient (large boxes versus a thin frame) and were directly related to the feedback information. That is, after making their selection, participants saw a small blue or green box which *primarily* informed them whether a blue or green choice had been rewarded on the current trial. From this information, the participant could, *secondarily*, infer whether the social information (red frame) was correct or incorrect.

Our *social-primary group* (n = 22; groups matched on age, gender, body mass index [BMI], and verbal working memory [VWM] span; *Table 1*) completed an adapted version of this task (*Figure 1B*) wherein the social information (red stimulus) was primary and the individual information (blue/green stimuli) was secondary. Participants first saw two placeholders; one empty and one containing a red box which indicated the social information. Subsequently, a thin green and a thin blue frame appeared around each placeholder. Participants were told that the red box represented the group's choice. They were then required to choose whether to go with the social group (red box) or not. After making their choice, a tick or cross appeared which *primarily* informed participants whether going with the

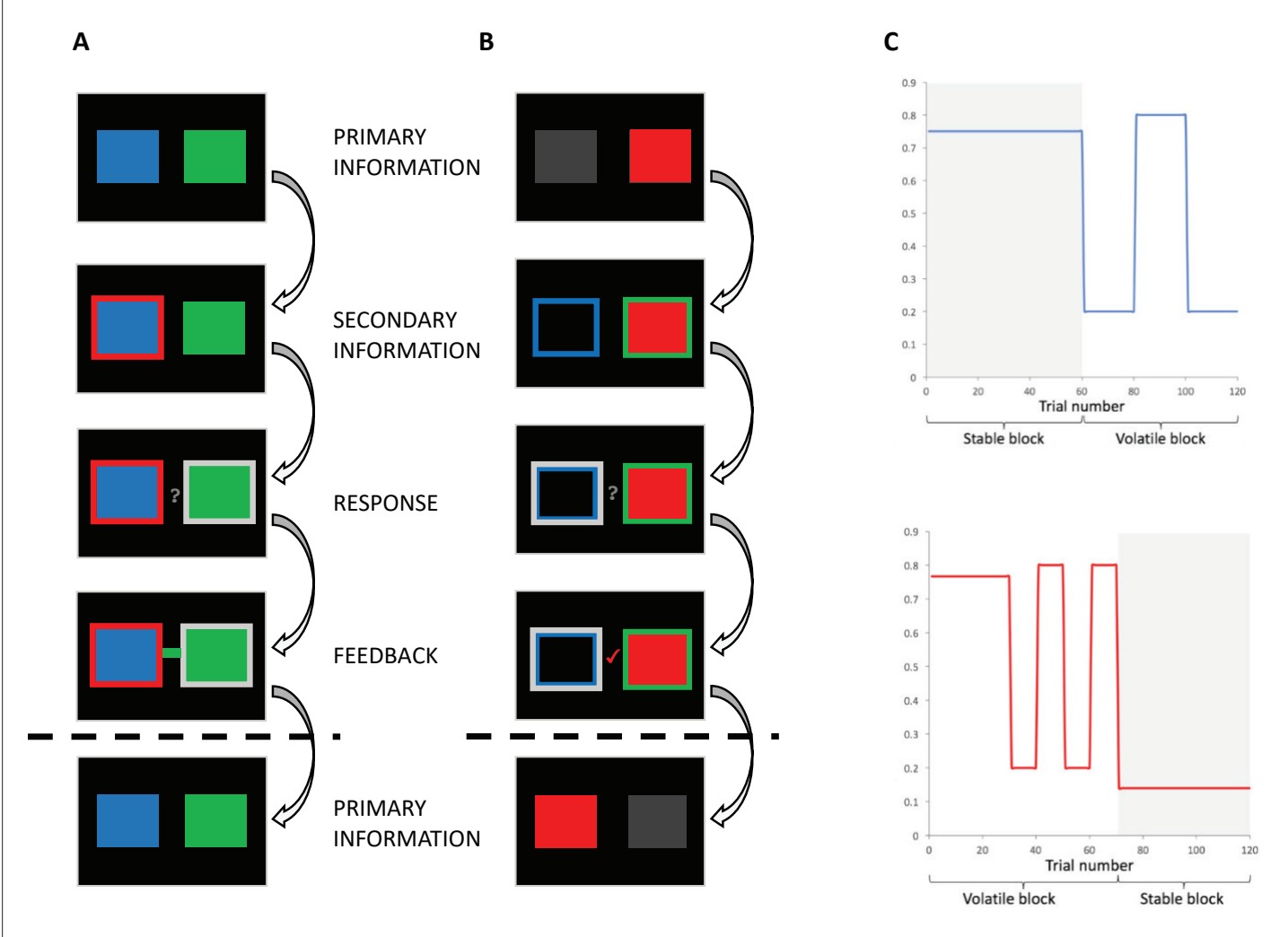

**Figure 1.** Behavioural task. (**A**) Individual-primary group. Participants selected between a blue and a green box to gain points. On each trial, the blue and green boxes were presented first. After 1-4 seconds (s), one of the boxes was highlighted with a red frame, representing the social information. After 0.5–2s, a question mark appeared, indicating that participants were able to make their response. Response was indicated by a silver frame surrounding their choice. After a 1-3s interval, participants received feedback in the form of a green or blue box in the middle of the screen. (**B**) Social-primary group. Participants selected between going with, or against a red box, which represented the social information. On each trial, the red box was displayed. After 1-4s, blue and green frames appeared. After 0.5–2s, a question mark appeared, indicating that participants were able to make their response. Response was indicated by a silver frame surrounding their choice. After a 1-3s interval, participants received feedback in the form of a tick or a cross. This feedback informed participants if going with the group was correct or incorrect, from this feedback participants could infer whether the blue or green frame was correct. (**C**) Example of pseudo-randomised probabilistic schedule. The probability of reward varied according to probabilistic schedules, including stable and volatile blocks for both the probability of the blue box/frame being correct (top) and the probability of the red (social) box/frame being correct (bottom).

social information was the correct option. From this they could, *secondarily*, infer whether the blue or green frame was correct. Consequently, for the social-primary group the social information was primary on the basis that it appeared first on the screen, highly salient (a large red box versus thin green/blue frames), and directly related to the feedback information.

Participants in both the individual-primary and social-primary groups performed 120 trials of the task on each of two separate study days. To perturb learning, on one day participants took 2.5 mg of HAL, previously shown to affect learning (*Pessiglione et al., 2006*) via multiple routes including perturbation of phasic dopamine signalling (*Schultz, 2007*; *Schultz et al., 1997*) facilitated by action at mesolimbic D2 receptors (*Camps et al., 1989*; *Grace, 2002*; *Lidow et al., 1991*). On the other day, they took a PLA under double-blind conditions, with the order of the days counterbalanced.

**Table 1.** Participant information.

| | Individual-primary group (n = 15) Mean (SD) | Social-primary group (n = 16) Mean (SD) | t (1,29) | $X^2$ (1, N = 31) | p-Value |
|---|---|---|---|---|---|
| Gender (n males: n females) | 7:8 | 8:8 | | 0.034 | 0.853 |
| Age | 25.600 (5.448) | 25.625 (4.745) | 0.014 | | 0.989 |
| VWM | 80.333 (6.016) | 76.354 (7.823) | 1.580 | | 0.125 |
| BMI | 24.016 (2.807) | 22.625 (2.606) | 1.431 | | 0.114 |

Age, gender, BMI, and VWM did not significantly differ between the groups.

SD: standard deviation; VWM: verbal working memory span; BMI: body mass index.

43 participants took part in at least one study day, 33 participants completed both study days. Two participants performed at below-chance-level accuracy and were excluded from further analysis. We present an analysis of data from the 31 participants who completed both study days with above-chance accuracy (*Table 1*) in this article, which we complement with a full analysis of all 41 datasets in Appendix 4i.

We used the following strategy to analyse our data. First, we sought to validate our manipulation by testing (under PLA) whether participants in both the individual-primary and social-primary groups learned in a more optimal fashion from the primary, versus secondary, source of information. Next, we tested our primary hypothesis that both social and individual learning would be modulated by HAL when they are the *primary* source of learning, but not when they comprise the *secondary* source. To do so, we estimated learning rates for primary and secondary sources of information, for each group (social-primary, individual-primary), under HAL and PLA, by fitting an adapted RW learning model to choice data. To ascertain that our model accurately described choices, we used simulations and parameter recovery. We used random-effects Bayesian model selection (BMS) to compare our model with alternative models. These analyses provided confidence that our model accurately described participants' behaviour. After testing our primary hypothesis, we explored the relationship between parameters from our computational model and performance. To accomplish this, we first used an optimal learner model, with the same architecture and priors as our adapted RW model, to assess the extent to which HAL made participants' learning rates more (or less) optimal. Finally, we regressed estimated model parameters against accuracy to gain insight into the extent to which variation in these parameters (and the effect of the drug thereupon) contributed to correct responses on the task.

## Social information is the primary source of learning for participants in the social-primary group

Our novel manipulation orthogonalised primary versus secondary and social versus individual learning. To validate our manipulation, we tested whether participants in both the individual-primary and social-primary group learned in a more optimal fashion from the primary versus secondary source of information in our PLA condition. For this validation analysis, we used a Bayesian learner model to create two optimal models: (1) an optimal primary learner and (2) an optimal secondary learner (Materials and methods). Subsequently, we regressed both models against participants' choice data, resulting in two $\beta_{optimal}$ values capturing the extent to which a participant made choices according to the optimal primary, and optimal secondary learner models, respectively. $\beta_{optimal}$ values were submitted to a repeated-measures analysis of variance (RM-ANOVA) with factors information source (primary, secondary) and group (social-primary, individual-primary), revealing main effects of information source (F(1,29) = 6.594, p=0.016) and group (F(1,29) = 10.423, p=0.003). $\beta_{optimal}$ values (averaged across individual-primary and social-primary groups) were significantly higher for the primary information ($\bar{x}(\sigma_{\bar{x}})$ = 0.872 (0.101)) compared with secondary information source ($\bar{x}(\sigma_{\bar{x}})$ = 0.438 (0.101); t(29) = 2.568, $p_{holm}$ = 0.016). $\beta_{optimal}$ values (averaged across primary and secondary conditions) were significantly higher for the social-primary group ($\bar{x}(\sigma_{\bar{x}})$ = 0.833 (0.078)) compared with the individual-primary group ($\bar{x}(\sigma_{\bar{x}})$ = 0.477 (0.078); t(29) = 3.228, $p_{holm}$ = 0.003) (*Figure 2*). Crucially, we did not observe a significant interaction between information and group (F(1,29) = 0.067, p=0.797), meaning that participants' choices were more influenced by the primary information source, regardless of whether

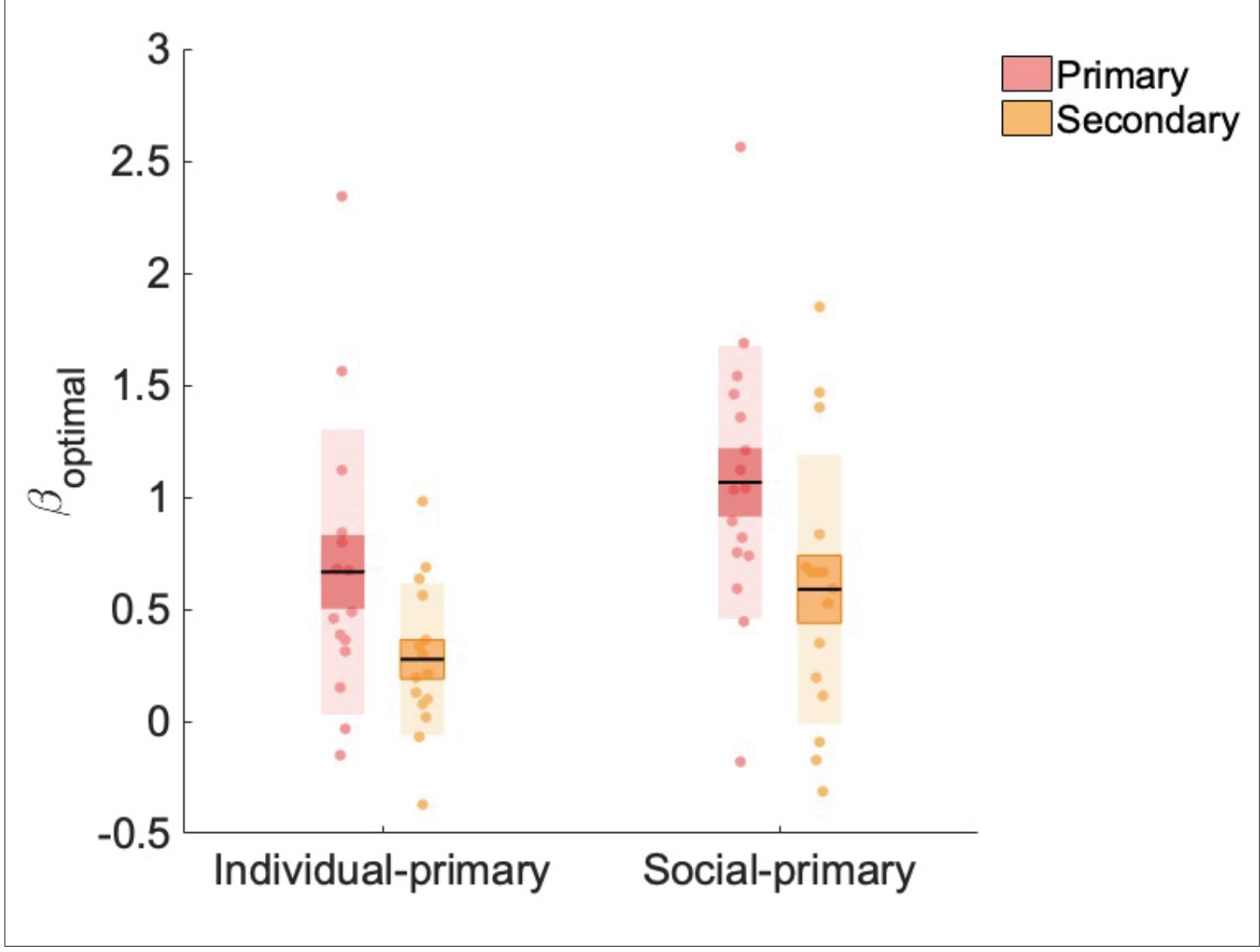

**Figure 2.** Beta weights (β_optimal) for primary and secondary information. $\beta_{optimal}$ values were significantly higher for the primary, compared to secondary, information source and for the social-primary, compared with the individual-primary, group. Data points indicate estimated $\beta\_optimal$ weights for individual participants (n = 31, placebo data only), bold point indicates the mean, and bold line indicates standard error of the mean (1 SEM).

it was social or individual in nature. Furthermore, $\beta_{optimal}$ values for primary information alone did not significantly differ between groups (t(29) = –1.982, $p_{holm}$ = 0.257). Note that $\beta_{optimal}$ weights for both information sources were significantly greater than zero (primary: t(30) = 7.534, p<0.001; secondary: t(30) = 4.789, p<0.001), thus our optimal models of information use explained a significant amount of variance in the use of both primary and secondary learning sources. These data show that, irrespective of social (or individual) nature, participants learned in a more optimal fashion from the primary (relative to secondary) learning source, which was first in the temporal order of events, highly salient and directly related to the reward feedback.

## Haloperidol reduces the rate of learning from primary sources

We hypothesised that both social and individual learning would be modulated by administration of the dopamine D2 receptor antagonist HAL when they were the *primary* source of learning, but not when they comprised the *secondary* source. To test this hypothesis, we fitted an adapted RW learning model (*Rescorla and Wagner, 1972*) to participants' choice data, enabling us to estimate various parameters that index learning from primary and secondary sources of information, for HAL and PLA conditions, for participants in the social-primary and individual-primary groups. Our adapted RW model provided

estimates, for each participant, of $\alpha$, $\beta$, and $\zeta$. The learning rate ($\alpha$) controls the weighting of prediction errors on each trial. A high $\alpha$ favours recent over (outdated) historical outcomes, while a low $\alpha$ suggests a more equal weighting of recent and more distant trials. Since our pseudo-random schedules included stable phases (where the reward probability associated with a particular option was constant for >30 trials), and volatile phases (where reward probabilities changed every 10–20 trials), $\alpha$ was estimated separately for volatile and stable phases (for both primary and secondary learning) to accord with previous research (*Behrens et al., 2007*; *Cook et al., 2019*; *Manning et al., 2017*). $\beta$ captures the extent to which learned probabilities determine choice, with a larger $\beta$ meaning that choices are more deterministic with regard to the learned probabilities. $\zeta$ represents the relative weighting of primary and secondary sources of information, with higher values indicating a bias towards the over-weighting of secondary relative to primary (see Materials and methods and Appendix 3 for further details of the model, model fitting, and model comparison).

We hypothesised an interaction between drug and (primary versus secondary) information source such that HAL would affect learning from the primary information source only, regardless of its social/individual nature. To test this hypothesis, we employed a linear mixed effects model with fixed factors information source (primary, secondary), drug (HAL, PLA), environmental volatility (volatile, stable), and group (social-primary, individual-primary) and dependent variable $\alpha$ (square-root transformed to

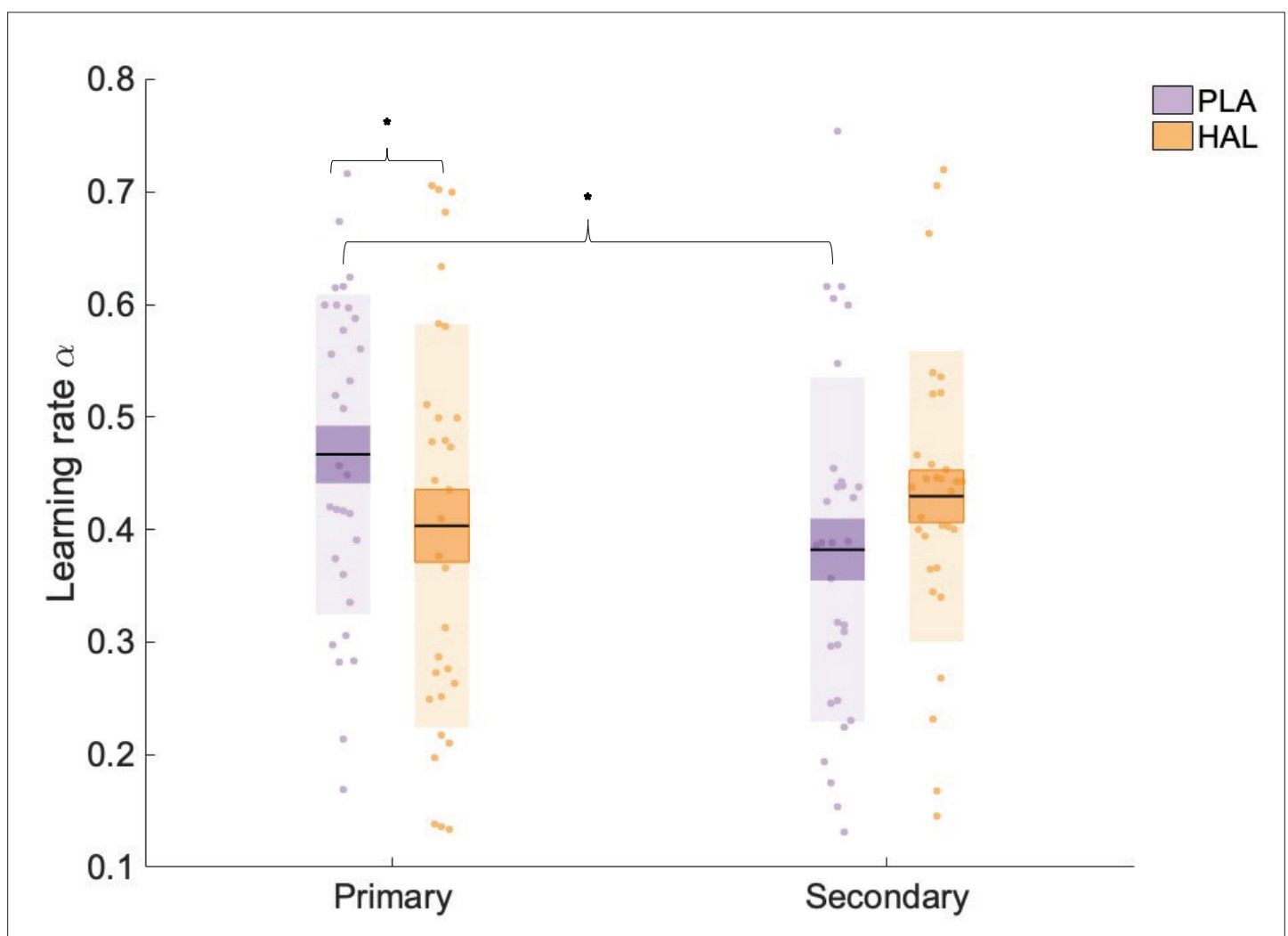

**Figure 3.** Learning rate ($\alpha$) estimates for learning from primary and secondary information across all trials (averaged across volatile and stable phases). There was a significant interaction between information and drug, with $\alpha$ estimates significantly lower under haloperidol (HAL; orange), relative to placebo (PLA; purple), for primary information only. Data points indicate square-root transformed $\alpha$ estimates for individual participants (n = 31). Boxes: standard error of the mean; shaded region: standard deviation. * indicates statistical significance (p<0.05).

meet assumptions of normality). We controlled for inter-individual differences by including random intercepts for subject. Including pseudo-randomisation schedule as a factor in all analyses did not change the pattern of results. The mixed model revealed a drug by information interaction (F(1, 203) = 6.852, p=0.009, beta estimate ($\sigma_{\bar{x}}$) = 0.026 (0.010), t = 2.62, confidence interval [CI] = [0.010–0.050]) (*Figure 3*). There were no significant main effects of drug (F(1, 203) = 0.074, p=0.786), group (F(1, 29) = 3.148, p=0.087), or volatility (F(1, 203) = 1.470, p=0.227) on $\alpha$ values, nor any other significant interactions involving drug (all p-values>0.05, see Appendix 4v–vi for analysis, including schedule, session, and working memory). Planned contrasts showed that, whilst under PLA, $\alpha_{primary}$ ($\bar{x}(\sigma_{\bar{x}})$ = 0.451 (0.025), collapsed across volatility and group) was significantly greater than $\alpha_{secondary}$ ($\bar{x}(\sigma_{\bar{x}})$ = 0.370 (0.025); z(30) = 2.861, p=0.004); this was not the case under HAL ($\alpha_{primary}$ $\bar{x}(\sigma_{\bar{x}})$ = 0.393 (0.025), $\alpha_{secondary}$ $\bar{x}(\sigma_{\bar{x}})$ = 0.417 (0.025); z(30) = –0.843, p=0.400). Furthermore, $\alpha_{primary}$ was decreased under HAL relative to PLA (z(30) = –2.050, p=0.040). Although $\alpha_{secondary}$ was, in contrast, numerically increased under HAL ($\bar{x}(\sigma_{\bar{x}})$ = 0.417 (0.025)) relative to PLA ($\bar{x}(\sigma_{\bar{x}})$ = 0.370 (0.025)), this difference was not significant (z(30) = 1.654, p=0.098). This drug × information interaction therefore illustrated that whilst HAL significantly reduced $\alpha_{primary}$ it had no significant effect on $\alpha_{secondary}$. Furthermore, under PLA there was a significant difference between $\alpha_{primary}$ and $\alpha_{secondary}$, which was nullified by HAL administration. Consequently, under PLA participants' rate of learning was typically higher for learning from the primary relative to the secondary source; however, under the D2 receptor antagonist HAL the rate of learning from the primary source was reduced and thus there was no significant difference in the rate of learning from primary and secondary sources.

Linear mixed models, with fixed factors group and drug, and random intercepts for subject, were also used to explore drug effects on $\zeta$ values (representing the relative weighting of primary/secondary information) and $\beta$ values. For $\zeta$, there were no significant main effects of drug (F(1, 29) = 1.941, p=0.174, beta estimate ($\sigma_{\bar{x}}$)= –0.07 (0.050), t = –1.390, CI = [–0.170 to 0.003]) or group (F(1, 51) = 0.184, p=0.669, beta estimate($\sigma_{\bar{x}}$)=0.020 (0.040), t = 0.430, CI = [–0.070 to 0.100]), nor drug by group interaction (F(1, 29) = 0.039, p=0.845, beta estimate($\sigma_{\bar{x}}$)=–0.001 (0.050), t = –0.200, CI = [-0.110 to 0.090]). Similarly, our analysis of $\beta$ values revealed no main/interaction effect(s) of drug, group, or drug by group (all p>0.05).

## Haloperidol reduces the rate of learning from a primary source irrespective of its social or individual nature

Our primary hypothesis was that HAL would modulate the rate of learning from the primary source irrespective of its social or individual nature. This would be evidenced as an interaction between drug and (primary versus secondary) information source (see above) in the absence of an interaction between drug, information source, and group (social-primary versus individual-primary). Crucially, we observed no significant interaction between drug, information source, and group (F(1, 203) = 0.098, p=0.754). To further assess whether drug effects on primary information differed as a function of group, results were also analysed within a Bayesian framework using JASP software (JASP Team 2020). A Bayes exclusion factor (BF$_{excl}$), representing the relative likelihood that a model without a drug × information × group interaction effect could best explain the observed data, was calculated (*Dienes, 2014*). Values of 3–10 are taken as moderate evidence in favour of the null hypotheses that there is no drug × information × group interaction (*Lee and Wagenmakers, 2013*) with values greater than 10 indicating strong evidence. The BF$_{excl}$ value was equal to 7.516, providing moderate evidence in favour of the null hypothesis that there is no drug × information × group interaction. Consequently, results confirmed our hypothesis: HAL perturbed learning from the primary but not the secondary source, irrespective of social or individual nature.

## Haloperidol brings $\alpha_{primary}$ estimates within the optimal range

To assess whether the effects of HAL on $\alpha_{primary}$ are harmful or beneficial with respect to performance, we first explored drug effects on accuracy (see Appendix 4ii for a detailed analysis including randomisation schedule). There was no significant difference in accuracy between HAL ($\bar{x}(\sigma_{\bar{x}})$ = 0.600 (0.013)) and PLA ($\bar{x}(\sigma_{\bar{x}})$ = 0.611 (0.010); F(1,29) = 0.904, p=0.349, $\eta_p^2$ = 0.030) conditions.

The lack of a significant main effect of drug on accuracy was somewhat surprising given the significant (interaction) effect on learning rates, that is, a decrease in $\alpha_{primary}$ under HAL relative to PLA. To

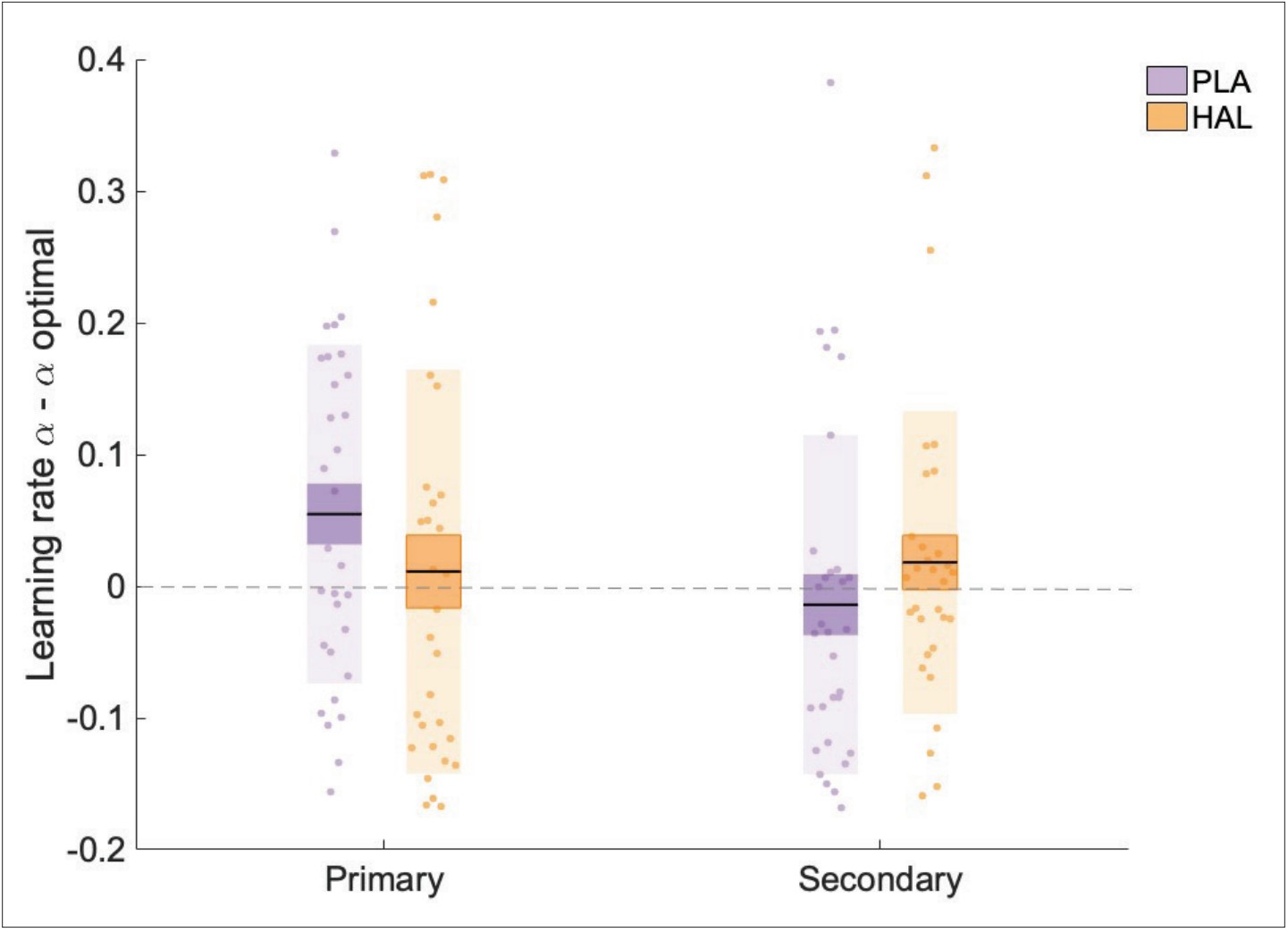

**Figure 4.** Learning rate estimates minus optimal learning rates. There was a significant interaction between information and drug, with $\alpha_{primary}$ scores significantly higher than optimal estimates under placebo (PLA) but not under haloperidol (HAL). Data points indicate $\alpha - \alpha_{optimal}$ values for individual participants (n = 31) across all trials (averaged across volatile and stable phases), Boxes: standard error of the mean; shaded region: standard deviation.

investigate whether HAL resulted in learning rates that were less, or alternatively, more, optimal, we compared our estimated $\alpha$ values with optimal $\alpha$ estimates. Since trial-wise outcomes were identical to those utilised by **Cook et al., 2019**, optimal values are also identical and are described here for completeness. An optimal learner model, with the same architecture and priors as the model employed in the current task, was fit to 100 synthetic datasets, resulting in average optimal learning rates: $\alpha_{optimal\_primary\_stable} = 0.16$, $\alpha_{optimal\_primary\_volatile} = 0.21$, $\alpha_{optimal\_secondary\_stable} = 0.17$, $\alpha_{optimal\_secondary\_volatile} = 0.19$. Scores representing the difference between (untransformed) $\alpha$ estimates and optimal $\alpha$ scores were calculated ($\alpha_{diff} = \alpha - \alpha_{optimal}$). A linear mixed model analysis on $\alpha_{diff}$ values with factors group, drug, volatility and information source, and random intercepts for subject was conducted. A significant interaction between drug and information source was observed (F(1, 203) = 4.895, p=0.028, beta estimate ($\sigma_{\bar{x}}$) = 0.019 (0.010), t = 2.212, CI = [0.000–0.040]) (**Figure 4**). Planned contrasts showed that, for primary information, $\alpha_{diff\_primary}$ was higher under PLA ($\bar{x}(\sigma_{\bar{x}}) = 0.052\ (0.023)$) compared with HAL ($\bar{x}(\sigma_{\bar{x}}) = 0.009\ (0.028)$); z(30) = 1.806, p=0.071. In contrast, $\alpha_{diff\_secondary}$ was lower under PLA ($\bar{x}(\sigma_{\bar{x}}) = -0.011\ (0.023)$) compared with HAL ($\bar{x}(\sigma_{\bar{x}}) = 0.021\ (0.021)$); z(30) = 1.323, p=0.186. Learning rates for learning from the primary source were higher than optimal under PLA, with $\alpha_{diff\_primary}$ significantly differing from 0 (one-sample t-test; t(30) = 2.259, p=0.031. HAL reduced learning rates that corresponded to learning from the primary source, thus bringing them within the optimal range, with $\alpha_{diff\_primary}$ not

significantly differing from 0 under HAL (one-sample $t$-test; t(30) = 0.319, p=0.752). Consequently, under HAL relative to PLA, learning rates for learning from primary sources were *more optimal*.

To explore whether α values were in some way related to accuracy scores, we used two separate backward regression models, for PLA and HAL conditions separately, with $α_{primary}$ and $α_{secondary}$ as predictors and accuracy as the dependent variable (see Appendix 4iii for details of a regression model with *all* model parameters). PLA accuracy was predicted by $α_{secondary}$ though this model only approached significance (R = 0.121, F(1,29) = 3.981, p=0.055). Under HAL, however, accuracy was predicted by a model with $α_{secondary}$ and $α_{primary}$ (R = 0.450, F(2,28) = 3.560, p=0.042), with $α_{primary}$ a significant positive predictor of accuracy (β = 0.404, p=0.028). Removing $α_{secondary}$ as a predictor did not significantly improve the fit of this model ($R^2$ change = 0.014, F change (1,29) = 0.495, p=1.000). When combined with our optimality analysis, these results suggest that under PLA $α_{primary}$ was outside of the optimal range of α values and thus accuracy was primarily driven by $α_{secondary}$. However, HAL reduced $α_{primary}$, bringing it within the optimal range. Thus, under HAL accuracy was driven by both $α_{primary}$ and $α_{secondary}$.

In sum, relative to PLA, the dopamine D2 receptor antagonist HAL significantly decreased learning rates relating to learning from primary, but not secondary sources of information, likely via mediation of phasic dopaminergic signalling (see Appendix 4iv). Interestingly, learning rates for learning from the primary source were higher than optimal under PLA and HAL brought them within the optimal range. Consequently, both primary and secondary learning contributed to accuracy under HAL but not under PLA. Importantly, the effects of HAL did not vary as a function of group allocation, which dictated whether the primary source was of social or individual nature. A Bayesian analysis confirmed that we had moderate evidence to support the conclusion that there was no interaction between drug, learning source and group. These data, thus, illustrate a dissociation along the primary-secondary but not social-individual axis.

## Discussion

This study tested the hypothesis that social and individual learning share common neurochemical mechanisms when they are matched in terms of (primary versus secondary) status. Specifically, we predicted that HAL would perturb learning from the primary but not the secondary source, irrespective of social or individual nature. Supporting our hypothesis, we observed an interaction between drug and information source (social versus individual) such that under HAL (compared to PLA) participants exhibited reduced learning rates with respect to learning from the primary, but not the secondary, source of information. Crucially, we did not observe an interaction between drug, information source, and group (social-primary versus individual-primary). Bayesian statistics revealed that, given the observed data, a model that excludes this interaction is 7.5 times more likely than models which include the interaction.

An important question concerns whether the lack of a dopaminergic dissociation between social and individual learning could be explained by participants not fully appreciating the social nature of the red shape (the social information source). In opposition to this, we argue that since our participants could not commence the task until reaching 100% accuracy in a pre-task quiz, which questioned participants about the social nature of the red shape, we can be confident that all participants knew that the red shape indicated information from previous participants. Participants also completed a post-task questionnaire (Appendix 5), which required them to reflect upon the extent to which their decisions were influenced by the social (red shape) and individual (blue/green shapes) information. If participants had not fully believed that the red shape represented social information, one might expect that they would indicate that they were *not* influenced by this source. In contrast, participants in both the individual-primary and social-primary groups believed that they were influenced by the red shape (as well as the blue/green stimuli). Furthermore, in our previous work, using the same social manipulation, we demonstrated that the personality trait social dominance significantly predicts social, but not individual, learning (*Cook et al., 2014*). Thus, illustrating that participants treat the social information differently from the non-social information in this type of paradigm. Finally, based on previous studies, we argue that even with a more overtly social manipulation it is highly likely that social learning would still be perturbed by dopaminergic modulation when social information is the primary source. Indeed, in a study by *Diaconescu et al., 2017* social information was represented by a video of a person indicating one of the two options. Even with this overtly social stimulus, Diaconescu

et al. still observed that social learning covaried with genetic polymorphisms that affect the functioning of the dopamine system.

The first part of our analysis illustrated that our manipulation produced the expected effect: when social information was first in the temporal order of events, highly salient and directly related to reward feedback participants learned in a more optimal fashion from this source of information. Such a result may be a surprise to some since one might think that, relative to learning from one's own experience, learning from others will always take a 'backseat'. Here, we clearly demonstrate that, when cast as the primary task, participants can make good use of social information. This paradigm may comprise a step towards developing a system to support accelerated social learning. Future studies could, for instance, investigate whether similar manipulations can be used to improve learning *about* (as opposed to *from*) other individuals. Since temporal order, saliency, and reward feedback were manipulated simultaneously, we cannot determine which manipulation is the most influential. Future work may therefore also seek to manipulate these factors independently to establish the most effective method for promoting social learning.

Our results comprise an important contribution to the debate concerning the existence of social-specific learning mechanisms. We find that, like individual learning, social learning is modulated by a dopaminergic manipulation when it is the primary source of information. This result marries well with previous studies that have linked social learning with dopamine-rich mechanisms when the social source has been the primary (or in many cases the sole) information source (*Campbell-Meiklejohn et al., 2010*; *Diaconescu et al., 2017*; *Klucharev et al., 2009*). Our results are also consistent with studies that have associated social learning with different neural correlates, outside of the dopamine-rich regions classically linked to individual learning, when it is a *secondary* source of information (*Behrens et al., 2008*; *Hill et al., 2016*; *Zhang and Gläscher, 2020*). Our data suggest that social and individual learning share common dopaminergic mechanisms when they are the primary learning source and that previous dissociations between these two learning types may be more appropriately thought of as dissociations between learning from a primary and secondary source. Extant studies (e.g., *Cook et al., 2019*) were not able to illustrate the importance of the primary versus secondary distinction because they did not fully orthogonalise primary versus secondary and social versus individual learning.

Though our results suggest shared neurochemical mechanisms for social and individual learning when they are matched in status, it is, nevertheless, essential to highlight that it does not follow that there are *no* dimensions along which social learning may be dissociated from individual learning. It is possible that although social and individual learning are affected by dopaminergic modulation – when they are the primary source – there are differences in the *location* of neural activity that could be revealed by neuroimaging. For instance, although social and individual learning are both associated with activity within the striatum (*Burke et al., 2010*; *Cooper et al., 2012*), social-specific activation patterns have been observed in other brain regions, including the temporoparietal junction (*Behrens et al., 2008*; *Lindström et al., 2018*) and the gyrus of the anterior cingulate cortex (*Behrens et al., 2008*; *Hill et al., 2016*; *Zhang and Gläscher, 2020*). Consequently, it is possible that HAL has comparable effects on social and individual learning but that these effects (seen at an 'algorithmic level of analysis', *Lockwood et al., 2020*) are associated with activity in different brain regions (i.e., dissociations at an 'implementation level of analysis', *Lockwood et al., 2020*). For example, HAL may comparably affect the BOLD signal associated with social and individual prediction errors, but the effect may be localised to dissociable neural pathways. Such a location-based dissociation requires further empirical investigation as well as further consideration of the possible functional significance of such location-based differences, if they are indeed present when primary versus secondary status is accounted for. Nevertheless, whilst such location-based differences are *possible*, we argue that they are not *probable* since, given different distributions of dopamine neurons, receptors, and reuptake mechanisms throughout the brain (*Grace, 2002*; *Korn et al., 2021*; *Matsumoto et al., 2003*; *Sulzer et al., 2016*), differences in location are relatively likely to result in differences in the magnitude of the effect of HAL (*Wächtler et al., 2020*; *Yael et al., 2013*). Additionally, since we did not observe significant effects of HAL on learning from social or individual sources when they were secondary in status, it remains a logical possibility that social and individual learning can be neurochemically dissociated when they are the secondary source of information – though it is admittedly difficult to conceive of a parsimonious explanation for the existence of two neurochemical mechanisms for

social and individual learning *from secondary sources.* Finally, it is possible that social and individual learning share common *dopaminergic* mechanisms when they are the primary source, but differentially recruit other neurochemical systems. For instance, some have argued that social learning may heavily rely upon serotonergic mechanisms (*Crişan et al., 2009*; *Frey and McCabe, 2020*; *Roberts et al., 2020*). The abovementioned avenues should be further explored; however, in the interim, it must be concluded that since existing studies have not controlled for primary versus secondary status, we do not currently have convincing evidence that social and individual learning can be dissociated in the human brain.

Notably, our results reveal a clear dissociation between learning from primary and secondary sources. For learning from primary sources HAL made learning rates more optimal, HAL did not have this effect on learning rates for secondary learning. Interestingly, a combined optimality analysis and regression model suggested that, under PLA, learning rates for learning from the primary source were 'too high' and fell outside of the optimal range (for this specific task). Consequently, under PLA, variance in accuracy was primarily explained by learning rates for learning from the secondary source. However, HAL reduced learning rates for learning from the primary source, bringing them within the optimal range. Thus, under HAL, accuracy was driven by learning rates for learning from both the primary and secondary sources. An open question concerns whether HAL truly *optimises* or simply *reduces* learning rate. Since the current paradigm was not designed to test this hypothesis, a reduction in learning rate herein also corresponds to an optimisation of learning rate. To dissociate the two, one would need a paradigm that generates sufficient numbers of participants with learning rates (in the PLA condition) that are *suboptimally low* such that one can observe whether, in these critical test cases, HAL *increases* (i.e., optimises) learning rate.

An intriguing question concerns the synaptic mechanisms by which HAL affects learning rates. Non-human animal studies have shown that phasic signalling of dopaminergic neurons in the mesolimbic pathway encodes reward prediction error signals (*Schultz, 2007*; *Schultz et al., 1997*). Since HAL has high affinity for D2 receptors (*Grace, 2002*), which are densely distributed in the mesolimbic pathway (*Camps et al., 1989*; *Lidow et al., 1991*), dopamine antagonists including HAL can affect phasic dopamine signals (*Frank and O'Reilly, 2006*) – either via binding at postsynaptic D2 receptors (which blocks the effects of phasic dopamine bursts) or via presynaptic autoreceptors (which has downstream effects on the release and reuptake of dopamine and thus modulates bursting itself) (*Benoit-Marand et al., 2001*; *Ford, 2014*; *Schmitz et al., 2003*). That is, HAL may affect learning rate via blockade of the postsynaptic D2 receptors, which may mute the effects of phasic dopamine signalling (either directly or via reduction in the background tonic rate of activity which, in turn, reduces the amplitude of phasic responses; *Belujon and Grace, 2015*; *Grace, 2016*), thus reducing the weight of prediction error signals on value updating (i.e., reducing the learning rate). Indeed, a number of studies have shown that HAL can attenuate prediction error-related signals (*Diederen et al., 2017*; *Haarsma et al., 2018*; *Menon et al., 2007*; *Pessiglione et al., 2006*). For example, in the context of individual learning, *Pessiglione et al., 2006* demonstrated that HAL attenuated prediction error signals in the striatum, indexed via changes in blood oxygen levels (BOLD). In addition to effects on postsynaptic D2 receptors, HAL may modulate prediction errors via effects on presynaptic autoreceptors. Autoreceptor binding is suggested to increase phasic bursting (*Dugast et al., 1997*; *Frank and O'Reilly, 2006*; *Garris et al., 2003*; *Pehek, 1999*), thus enhancing the phasic signal that is indicative of positive prediction errors. A combination of pre- and postsynaptic effects could feasibly result in more optimal learning rates wherein dopamine signalling is muted via postsynaptic blockade, thus muting (tonic background) 'noise' (and signal) but where the phasic 'signal' is enhanced via presynaptic effects, potentially resulting in an overall increased signal-to-noise ratio which may translate into more optimal learning rates.

Perhaps the most novel contribution of our work is that we here illustrate that, whilst dopaminergic modulation affects learning from the primary source, it does not significantly affect learning from the secondary source. Previous studies have illustrated that humans can learn – ostensibly simultaneously – from multiple sources of information and tend to organise this information in a hierarchical fashion such that the source which is currently of highest value has the greatest influence on a learner's behaviour (*Daw et al., 2006*). Here, we extend this work by showing that the primary source, at the top of the hierarchy, is more heavily influenced by modulation of the dopamine system, thus suggesting a graded involvement of the dopamine system according to a source's status in the 'learning hierarchy'.

Extant studies (*Daw et al., 2006*) suggest that such learning hierarchies are flexible and can be rapidly remodelled according to a source's current value. The success of our orthogonalisation of social versus individual and primary versus secondary learning depended on a within-subjects design, wherein the status (primary or secondary) of the learning source varied only between participants. Although our study was therefore not optimised for studying the rapid remodelling of learning hierarchies, our results pave the way for future studies to investigate whether the impact of dopaminergic modulation of learning from a particular source quickly changes according to the source's current status in the learning hierarchy.

In sum, in previous paradigms that dissociate social and individual learning, the social information comprised a secondary or additional information source, differing from individual information both in terms of its social nature (social/individual) and status (secondary/primary). We here provide evidence that dissociable effects of dopaminergic manipulation on different learning types are better explained by primary versus secondary status, than by social versus individual nature. Specifically, we showed that, relative to PLA, HAL reduced learning rates relating to learning from the primary, but not secondary, source of information irrespective of social versus individual nature. Results illustrate that social and individual learning share a common dependence on dopaminergic mechanisms when they are the primary learning source.

## Materials and methods

### Subjects

Subjects (n = 43, aged 19–42 years, mean [SD] = 26 (6.3); 19 female) were recruited from the University of Birmingham and surrounding areas in Birmingham city, via posters, email lists, and social media. Four participants dropped out of the study after completing the first day. A further five participants could not complete the second test day due to university-wide closures and a restriction of data collection. In total, 43 participants completed one session, with 33 participants completing both test days. However, Bayes exclusion factors were reported for interactions of interest to avoid the possibility of type 2 error. The study was in line with the local ethical guidelines approved by the local ethics committee (ERN_18_1588) and in accordance with the Helsinki Declaration of 1975.

### General procedure

The study protocol was pre-registered (see Open Science Framework [OSF], https://osf.io/drmjb, for study design and a priori sample size calculations). All participants attended a preliminary health screening session with a qualified clinician, followed by two test sessions with an interval of 1 to a maximum of 4 weeks between testing session. The health screening session, lasting approximately 1 hr, started with informed consent, followed by a medical screening. Participants were excluded from further participation if they met any of the exclusion criteria. Participants then completed a battery of validated questionnaire measures (see Appendix 1 for inclusion/exclusion criteria, questionnaire measures, medical symptoms, and mood ratings). Both test days (1–4 weeks post health screening) followed the same procedure, starting with informed consent, followed by a medical screening. Participants were then administered capsules (by a member of staff not involved in data collection) containing either 2.5 mg HAL or PLA in a double-blind, PLA-controlled, crossover design. Participants were told to abstain from alcohol and recreational drugs in the 24 hr prior to testing and from eating in the 2 hr prior to capsule intake.

1.5 hr after capsule intake, participants commenced a battery of behavioural tasks, including a probabilistic learning paradigm (Go-No-Go learning; *Frank and O'Reilly, 2006*) and a measure of VWM (*Sternberg, 1969*). The social learning task was started approximately 3 hr post-capsule administration, within the peak of HAL blood plasma concentration. HAL dosage and administration times were in line with similar studies which demonstrated both behavioural and psychological effects of HAL (*Bestmann et al., 2014*; *Frank and O'Reilly, 2006*). Both test days lasted approximately 5.5 hr in total. Blood pressure, mood, and medical symptoms were monitored throughout each day: before capsule intake, three times during the task battery and after finishing the task battery. On completion of the second session, participants reported on which day they thought they had taken the active drug or PLA. Participants received monetary compensation on completion of both testing sessions,

at a rate of £10 per hour, with the opportunity to add an additional £5 based on their performance during the learning task.

## Behavioural task

Participants completed a modified version of a social learning task (*Cook et al., 2014*), first developed by *Behrens et al., 2008*. The task was programmed using MATLAB R2017b (The MathWorks, Natick, MA). Participants were randomly allocated to one of two groups. For both groups, participants completed 120 trials on both test days. The task lasted approximately 35 min, including instructions. Before the main task, participants completed a step-by-step on-screen practice task (10 trials) in which they learnt to choose between the two options to obtain a reward and learned that the 'advice' represented by the frame(s) could help in making the correct choice in some phases. In our previous work with the individual-primary condition alone, we demonstrated that social dominance significantly predicts social, but not individual, learning (*Cook et al., 2014*). Thus, showing that participants maintain a conceptual distinction between the social and individual learning sources. In this study, we investigated whether participants maintained this conceptual distinction by requiring participants to complete a short quiz (three questions), testing their knowledge, after the practice task. Participants were required to repeat the practice round until they achieved 100% correct score in the quiz, meaning that all participants understood the structure of the task, and that the red shape represented *social* information. Furthermore, after the experiment, participants completed a feedback questionnaire (Appendix 5). Answers confirmed that participants understood the difference between, and paid attention to both, individual and social sources of information. Participants were informed as to whether they had earned a £5 bonus after the second session. Due to ethical considerations, all participants received the bonus.

## Individual-primary group

On each trial, participants were required to choose between a blue or green box to gain points. Participants could also use an additional, secondary, source of information – a red frame surrounding either the blue or green box – to help make their decision. Participants were informed (see Appendix 5 for instruction scripts) that the frame represented the most popular choice made by a group of participants who had previously completed the task. They were also informed that the task followed 'phases' wherein sometimes the blue, but at other times the green choice, was more likely to result in reward and sometimes the social information predominantly indicated the correct box, but at other times it predominantly surrounded the incorrect box (*Figure 1A*). After making their choice, participants received outcome information in the form of a blue or green indicator. The indicator primarily informed participants about whether the blue or green box had been rewarded on the current trial. Whether the social information surrounded the correct or incorrect box could, secondarily, be inferred from the indicator. For example, if the red frame indicated that the social group had chosen the blue shape, and the blue shape was shown to be correct, participants could infer that the social information had therefore been correct on that trial. Both the probability of reward associated with the blue/green stimuli and the utility of the social information varied according to separate probabilistic schedules, with participants randomly assigned to one of four groups (Appendix 2). For both individual and social information, the probabilistic schedules featured stable phases, where the probability of reward was constant, and volatile phases, in which the probability switched every 10–20 trials. This feature of the task design was included to capture potential effects of dopaminergic modulation on adaptation to environmental volatility (*Cook et al., 2019*). Participants were informed that correct choices would be rewarded, and thus to aim to accumulate points to obtain a reward at the end of the experiment. Although probabilistic schedules for day 2 were the same as day 1, there was variation in the trial-by-trial outcomes and advice. In addition, to prevent participants from transferring learned stimulus-reward associations from day 1 to day 2, different coloured stimuli were employed on the second session: participants viewed blue/green squares with advice represented as a red frame on day 1 and yellow/purple squares with advice represented as a blue frame on day 2.

## Social-primary group

For the social-primary group, the social information source was the primary source of learning. On each trial, participants were presented with two grey placeholders. One placeholder was filled with

**Table 2.** Untransformed estimated learning rates.

| | | $\alpha_{primary\_volatile}$ | $\alpha_{primary\_stable}$ | $\alpha_{secondary\_volatile}$ | $\alpha_{secondary\_stable}$ |
|---|---|---|---|---|---|
| PLA | $\bar{x}(\sigma_{\bar{x}})$ | 0.184 (0.018) | 0.290 (0.041) | 0.187 (0.028) | 0.151 (0.025) |
| | Range | 0.024–0.477 | 0.027–0.721 | 0.011–0.591 | 0.004–0.612 |
| HAL | $\bar{x}(\sigma_{\bar{x}})$ | 0.169 (0.029) | 0.218 (0.033) | 0.200 (0.023) | 0.202 (0.026) |
| | Range | 0.010–0.578 | 0.013–0.699 | 0.014–0.481 | 0.011–0.584 |

$\bar{x}(\sigma_{\bar{x}})$: mean (standard error of the mean); PLA: placebo; HAL: haloperidol.

a red box, indicating the group's choice. Blue/green frames then appeared around the placeholders. As in the individual-primary group, participants were informed that the task followed 'phases' wherein sometimes going with, but at other times going against, the group's choice was more likely to result in reward and sometimes the blue frame predominantly indicated the correct box, whereas at other times the green frame predominantly indicated the correct box. After making their choice, participants received outcome information in the form of a tick/cross indicator. The indicator primarily informed participants about whether the social group had been rewarded (and thus going with them would have resulted in points scoring but going against them would not) on the current trial. Whether the blue (green) frame surrounded the correct or incorrect option could, secondarily, be inferred from the indicator. As in the individual-primary task, both the probability of reward associated with the blue/green stimuli and the utility of the social information varied according to probabilistic schedules (Appendix 2). All other aspects of the task structure were the same as previously described in the individual-primary task group.

## Data analysis

All analyses were conducted using MATLAB R2017b (The MathWorks) and Bayesian analyses using JASP (JASP Team, 2020, JASP, [version 0.14, computer software]). Linear mixed models were fitted to data using RStudio (RStudio Team, 2020, RStudio: Integrated Development for R, RStudio, PBC, Boston, MA). In the instance of data not meeting assumptions of normality (as assessed by Kolmogorov–Smirnov testing), data were square-root-transformed. Learning rate $\alpha$ values were square-root transformed (see *Table 2* for untransformed learning rates). We used the standard p<0.05 criteria for determining if significant effects were observed, with a Holm correction applied for unplanned multiple comparisons, to control for type I family-wise errors. In addition, effect sizes and beta weights for linear mixed model analysis are reported.

## Data preprocessing

Datasets were excluded based on the following: accuracy < 50% under PLA, chose the same side (left/right) or colour on >80% trials, and incomplete datasets (less than 120 trials completed). Two subjects were excluded, resulting in a final sample of n = 31, with behavioural data for both testing days, and n = 41, with data for 1 day only (see Appendix 4i for analysis).

## Optimal learner model

The influence of each information source (primary and secondary) on choices was quantified by regressing two 'optimal learners' against subjects' choices. The first comprised an optimal 'individual learner model', which was generated by using a Bayesian learner algorithm (*Behrens et al., 2007*), to simulate an optimal learner who learns solely from individual information (the blue and green stimuli). The second comprised a 'social learner model' which simulated an optimal learner who learns solely from the social information (red stimuli). The Bayesian learner algorithm (*Behrens et al., 2007*) describes an optimal approach to tracking reward probabilities in a changing environment. It assumes an underlying probability of an outcome being correct and tracks this probability across time, as well as maintaining an estimate of the rate of change of probabilities, that is, volatility. All probabilities are updated in a Markovian fashion, meaning that there is no requirement to store the full history of decision outcomes or statistics of the environment (*Behrens et al., 2007*). Thus, on each trial, the individual learner model represented the reward probability associated with a blue choice,

derived through learning, in an optimal fashion, exclusively from information about reward outcomes and ignoring the social information. The social learner model represented the probability, based on the (reward-weighted) social information, that the social information was correct. From the social learner model, on each trial, the reward probability of a blue choice was calculated, which would have been derived if a participant had been learning optimally, exclusively from the social information (i.e., ignoring individual reward outcomes). Subsequently both models were regressed separately against each individual participant's choice data using binomial logistic regression, with model predictions from the primary and secondary models as continuous predictor variables and participant response as the dependent variable (0/1). For each participant, this produced two parameter estimates, or standardised beta weights, each representing the degree to which individual experience and social information explained choices. For example, a participant whose choices were more strongly influenced by the social information than the individual information would have a high social $\beta_{optimal}$ value, and a low individual $\beta_{optimal}$ value.

## Computational modelling framework

Participant responses were modelled using an adapted RW learning model (*Rescorla and Wagner, 1972*). The model relies on the assumption that updates to choice behaviour are based on prediction errors, that is, the difference between an expected and the actual outcome. Participants were assumed to update their beliefs about outcomes based on sensory feedback (perceptual model) and to use this feedback to make decisions about the next action (response model). Model fitting was performed using scripts adapted from the TAPAS toolbox (*Diaconescu et al., 2014*; scripts available at OSF link; https://tinyurl.com/b3c7d2zb). A systematic comparison of eight separate models (see Appendix 3 for full details regarding model fitting and model comparison) showed that the exceedance probability of this particular model was ~1. This demonstrates (relative) evidence in favour of the conclusion that, the current model, with separate learning rates for primary and secondary information, and volatile and stable phases, provided the best fit to participant choice data and that the data likely originated from the same model for both HAL and PLA treatment conditions (*Appendix 3—figure 1*). Further model validation, including simulation of data and parameter recovery, provided further support for the choice of computational model (Appendix 3).

## Perceptual model

The RW predictors used in our learning models consisted of a modified version of a simple learning model, with one free parameter, the learning rate $\alpha$, varying between 0 and 1.

$$V_{(i+1)} = V_i + \alpha \left( r_i - V_i \right)$$

According to this model, the predicted value ($V_i$) is updated on each trial based on the prediction error (PE), or the difference between the actual and the expected reward $\left( r_i - V_i \right)$, weighted by the learning rate $\alpha$. $\alpha$ thus captures the extent to which the PE updates the estimated value on the next trial. In line with previous work (*Cook et al., 2019*), we used an extended version of this learning model, with separate $\alpha$ values for volatile and stable environmental phases. In a stable environment, learning rate will optimally be low, and reward outcomes over many trials will be taken into account. In a volatile environment, however, an increased learning rate is optimal as more recent trials are used to update choice behaviour (*Behrens et al., 2007*). Furthermore, we simultaneously ran two RW predictors in order to estimate parameters relating to learning from primary and secondary information sources. Consequently, our model generated the predicted value of going with the primary source (going with the blue frame for the individual-primary group, going with the group for the social-primary group; $V_{\_primary(i+1)}$) and the predicted value of the secondary information (going with the group recommendation for the individual-primary group, going with the blue frame for the social-primary group; $V_{\_secondary(i+1)}$) and provided four α estimates: $\alpha_{primary\_stable}$, $\alpha_{primary\_volatile}$, $\alpha_{secondary\_stable}$, and $\alpha_{secondary\_volatile}$.

## Response model

Our response model assumed that participants integrated learning from both primary and secondary sources. The action selector predicts the probability that the primary information (blue choice/ group choice) will be rewarded on a given trial and was based on the softmax function (TAPAS toolbox), adapted by *Diaconescu et al., 2014*. This response model was adapted from that used by *Cook*

*et al., 2019* and is reproduced here with permission. The value of primary and secondary information was combined using the following:

$$V_{\_primary(i+1)} = \zeta(V_{\_secondary\_advice\_weighted(i+1)}) + (1-\zeta)(V_{\_primary(i+1)})$$

where $\zeta$ is a parameter that varies between individuals and that controls the weighting of secondary relative to primary sources of information. $V_{\_secondary\_advice\_weighted(i+1)}$ comprises the advice provided by the secondary information (the red and blue frames, for individual-primary and social-primary groups, respectively) weighted by the probability of advice accuracy ($V_{\_secondary(i+1)}$) in the context of making a choice to go with the primary information (the blue and red box for the individual-primary and social-primary groups, respectively). That is,

$$V_{\_secondary\_advice\_weighted(i+1)} = \left| \text{advice} - V_{\_secondary(i+1)} \right|$$

where advice from the red frame equals 0 for blue and 1 for green, and advice from the blue frame equals 0 for going with the red box and 1 for going against the red box. For example, for a participant in the social-primary group, if the blue frame advised them to go with the red box (the group choice) and the probability of advice accuracy was estimated at 80% ($V_{\_secondary(i+1)} = 0.80$), the probability that the choice to go with the group will be rewarded, inferred from secondary learning, would be 0.8 ($V_{\_secondary\_advice\_weighted(i+1)} = |0–0.8| = 0.8$). The probability that this integrated belief would determine participant choice was described by a unit square sigmoid function, describing how learned belief values are translated into choices.

$$P\left(y_{(i+1)} = 1 \| V_{\_primary(i+1)}\right) = \frac{V_{\_primary(i+1)}{}^{\beta}}{V_{\_primary(i+1)}{}^{\beta} + \left(1 - V_{\_primary(i+1)}\right)^{\beta}}$$

Here, responses are coded as $y_{(i+1)} = 1$ when selecting the primary option (going with the blue and red box for the individual-primary and social-primary groups, respectively), and $y_{(i+1)} = 0$ when selecting the alternative (going with the green box and going against the red box for the individual-primary and social-primary groups, respectively). The participant-specific free parameter $\beta$, the inverse of the decision temperature, describes the extent to which the estimated value of choices determines actual participant choice: as $\beta$ decreases, decision noise increases and decisions become more stochastic; as $\beta$ increases, decisions become more deterministic towards the higher value option.

## Significance tests for estimated model parameters

Parameters were fitted separately for each participant's choice data. Learning rate ($\alpha$) was estimated for each participant, primary and secondary learning, and volatile and stable phases, on both test days, resulting in eight estimated learning rates per participant. $\beta$ values were also estimated for each participant on both treatment days, resulting in two $\beta$ values per participant. Effects-coded mixed model linear analyses were carried out to allow for inclusion of subject as a random factor, thus ensuring that between-participant variation in $\alpha$ could be controlled for. Fixed factors were drug (HAL, PLA), information type (primary, secondary), volatility (volatile, stable), and group (individual-primary, social-primary), with the inclusion of random intercepts for participant: ~group × information × drug × volatility +1| subject.

RM-ANOVA for linear mixed effects models was carried out using the Satterthwaite approximation for degrees of freedom, and the model was fit using maximum likelihood estimation, with a model including random intercepts, but not random slopes, providing the best fit to the data. All analyses were repeated with and without the inclusion of age, BMI, and baseline working memory as covariates, with the pattern of results unchanged. Where appropriate, data were transformed to meet assumptions of normality for parametric testing.

## Bayesian statistical testing

Bayesian statistical testing was implemented as a supplement to null hypothesis significance tests to investigate if null results represent a true lack of a difference between the groups (*Dienes, 2014*) using JASP software based on the R package 'BayesFactor' (*Rouder et al., 2012*). The JASP framework for RM-ANOVA was used (*van den Bergh et al., 2020*), whereby exclusion Bayes factors were

obtained for predictors of interest. The exclusion Bayes factor ($BF_{excl}$) for a given predictor or interaction quantifies the change in odds from the prior probability that the predictor is included in the regression model to the probability of exclusion in the model after seeing the data ($BF_{excl}$). Bayes factors were computed by comparing all models with a predictor against all models without that predictor, that is, comparing models that contain the effect of interest to equivalent models stripped of the effect. For example, an exclusion Bayes factor for an effect of 3 for a given predictor i can be interpreted as stating that models which exclude the predictor i are three times more likely to describe the observed data than models which include the predictor. In short, the exclusion Bayes factor is interpreted as the evidence given the observed data for excluding a certain predictor in the model and can be used as evidence to support null results. For all Bayesian analyses, the Bayes factor quantifies the relative evidence for one theory or model over another. We followed the classification scheme used in JASP (*Lee and Wagenmakers, 2013*) to classify the strength of evidence given by the Bayes factors, with $BF_{excl}$ between 1 and 3 considered as weak evidence, between 3 and 10 as moderate evidence, and greater than 10 as strong evidence for the alternative hypothesis, respectively.

## Acknowledgements

We acknowledge Ms Lydia Hickman for assistance with data collection and Dr Kasim Qureshi and Dr Hannah Liu for medical screening. AJR's role in this project was supported by a Midlands Integrative Biosciences Training Partnership (MIBTP) – Biotechnology and Biological Sciences Research Council (BBSRC) PhD studentship. JLC, SLS, and BS were supported by the European Union's Horizon 2020 Research and Innovation Programme under European Research Council (ERC)-2017-STG Grant Agreement No. 757583 (Brain2Bee).

## Additional information

### Funding

| Funder | Grant reference number | Author |
|---|---|---|
| Biotechnology and Biological Sciences Research Council | Midlands Integrative Biosciences Training Partnership (MIBTP) Doctoral Funding | Alicia J Rybicki |
| H2020 European Research Council | 757583 - Brain2Bee | Sophie L Sowden Bianca Schuster Jennifer L Cook |

The funders had no role in study design, data collection and interpretation, or the decision to submit the work for publication.

### Author contributions

Alicia J Rybicki, Data curation, Formal analysis, Investigation, Methodology, Project administration, Resources, Software, Validation, Visualization, Writing – original draft, Writing – review and editing; Sophie L Sowden, Investigation, Project administration; Bianca Schuster, Investigation; Jennifer L Cook, Conceptualization, Funding acquisition, Methodology, Project administration, Resources, Software, Supervision, Writing – review and editing

### Author ORCIDs

Alicia J Rybicki http://orcid.org/0000-0001-6668-1214
Sophie L Sowden http://orcid.org/0000-0001-9913-0515
Jennifer L Cook http://orcid.org/0000-0003-4916-8667

### Ethics

Human subjects: Informed consent was obtained from each subject. The study was in line with the local ethical guidelines approved by the local ethics committee (ERN_18_1588) and in accordance with the Helsinki Declaration of 1975.

**Decision letter and Author response**

Decision letter https://doi.org/10.7554/eLife.74893.sa1

Author response https://doi.org/10.7554/eLife.74893.sa2

---

## Additional files

### Supplementary files
• Transparent reporting form

### Data availability
All raw data and analysis scripts can be accessed at the Open Science Framework data repository.

The following dataset was generated:

| Author(s) | Year | Dataset title | Dataset URL | Database and Identifier |
|-----------|------|---------------|-------------|-------------------------|
| Rybicki A, Cook J | 2021 | Dopaminergic challenge dissociates learning from primary versus secondary sources of information | https://osf.io/398w4/ | Open Science Framework, 398w4 |

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

## Appendix 1

### Inclusion criteria

Participant is willing and able to give informed consent for participation in the study.
Aged between 18 and 45.
BMI in the range of 18.5–29.5.
Resting blood pressure in the range of 90/60 (low) to 140/90 (high).
Electrocardiogram QT (heart rate corrected) interval <0.42.

### Exclusion criteria

Participation in another drug study in the 3 weeks previous.
Personal or first-degree family history of cardiovascular disease, specifically hypotension, arrhythmias or valvular disease, stroke.
Neurological abnormalities or traumas, kidney disease, or liver disease.
Inherited blood conditions.
Psychiatric or psychological conditions (including depression and anxiety disorders).
Known learning disability.
Anybody found to have an elongated Q-T interval following single lead ECG examination.
Low heart rate.
Low or high blood pressure.
Any regular medication – excluding the oral contraceptive pill.
Recent recreational drugs use or alcohol and drug dependency.
Known allergy to any medication.
Current pregnancy or breastfeeding.
Previous participant in a drug study.
Lack of sleep in last 24 hr.
Lack of food or drink in last 12 hr.
Primary sensory impairment (e.g., uncorrected visual or hearing impairment).
Lactose intolerant.
Insufficient English to be able to consent to take part in the study.

### Baseline cognitive measures and mood ratings

Approximately 1 week prior to drug/PLA administration, participants completed a battery of self-report questionnaire measures: Autism Spectrum quotient (AQ) (Baron-Cohen et al., 2001), Toronto Alexithymia Scale (TAS 20) (*Bagby et al., 1994*), Behavioural Inhibition/Activation Scale (BIS-BAS) (*Carver and White, 1994*), the Depression Anxiety and Stress Scale (DASS 21) (*Lovibond and Lovibond, 1995*), Interpersonal Reactivity Index (IRI) (*Davis, 1983*), Beck's Depression Inventory (BDI) (*Beck et al., 1996*), and Body Perception Questionnaire (BPQ) (*Porges, 1993*). Self-report questionnaire scores are summarised in *Table 1*. The individual-primary group did not differ significantly on any measure from the social-primary group. The group that received HAL on day 1 did not differ significantly on any of the baseline measures from the group that received PLA on day 1 (p<0.05). Mood and fatigue were monitored three times per day during each test day (1) before capsule intake, (2) 2 hr post-capsule intake upon start task battery, and (3) upon completion of the task battery. The mood ratings consisted of the Positive and Negative Affect Scale (PANAS) (*Watson et al., 1988*). A self-report scale was used to monitor fatigue. 24% of participants reported that they did not know on which day they had taken an active drug. Out of the remaining participants, 84% of participants correctly reported that they thought they had received an active drug. No adverse side effects were reported. Blood pressure, heart rate, and blood oxygenation levels were monitored five times over the course of the testing days; before drug/PLA administration, and then at 1, 2, and 3 and a half hour intervals thereafter. Measures were taken for a final time immediately before the end of the testing day.

**Appendix 1—table 1.** Self-report questionnaire scores (n = 31).

| Self-report questionnaires | Individual-primary group | Social-primary group | t (29) | p-Value |
|---|---|---|---|---|
| AQ | 9.412 (4.556) | 6.500 (4.179) | 1.910 | 0.065 |
| TAS-20 | 39.529 (6.947) | 40.313 (7.981) | −0.301 | 0.765 |
| BIS-BAS | 50.647 (6.855) | 51.125 (5.536) | −0.219 | 0.828 |
| DASS-Stress | 3.176 (4.231) | 3.875 (2.306) | −0.583 | 0.723 |
| DASS-Anxiety | 1.353 (2.178) | 1.938 (2.516) | −0.715 | 0.564 |
| DASS-Depression | 1.706 (1.863) | 2.313 (3.005) | −0.702 | 0.480 |
| IRI | 66.235 (15.114) | 66.375 (10.645) | −0.031 | 0.976 |
| BDI | 3.176 (3.746) | 3.438 (2.732) | −0.227 | 0.822 |
| BPQ | 52.176 (29.473) | 46.688 (18.650) | 0.635 | 0.221 |

Mean (standard deviation) scores are reported. Significance level for the between-group differences are reported. Autism Spectrum quotient (AQ) (**Baron-Cohen et al., 2001**), Toronto Alexithymia Scale (TAS 20) (**Bagby et al., 1994**), Behavioural Inhibition/Activation Scale (BIS-BAS) (**Carver and White, 1994**), the Depression Anxiety and Stress Scale (DASS 21) (**Lovibond and Lovibond, 1995**), Interpersonal Reactivity Index (IRI) (**Davis, 1983**), Beck's Depression Inventory (BDI) (**Beck et al., 1996**), and Body Perception Questionnaire (BPQ) (**Porges, 1993**).

## Drug effects on mood and tiredness

PANAS scores were submitted to separate RM-ANOVAs, with within-subjects (WS) factors time (baseline/start testing/end testing) and drug (HAL/PLA). For both positive and negative scores, a main effect of time was observed. Both positive ($F_{(2,62)} = 8.286$, $p<0.001$, $\eta_p^2 = 0.211$) and negative scores decreased over time ($F_{(2,62)} = 6.020$, $p=0.004$, $\eta_p^2 = 0.163$). A drug by time interaction was observed for positive scores ($F_{(2,62)} = 7.353$, $p=0.001$, $\eta_p^2 = 0.192$), with simple effects analysis demonstrating that positive scores decreased over time under HAL ($p<0.001$), but not PLA ($p=0.994$). A main effect of drug was observed on negative scores ($F_{(1,31)} = 4.749$, $p=0.037$, $\eta_p^2 = 0.133$), with higher negative affect scores under HAL ($\bar{x}(\sigma_{\bar{x}}) = 10.771$ (0.557)) compared with PLA ($\bar{x}(\sigma_{\bar{x}}) = 9.491$ (0.557)).

Self-reported fatigue ratings (Likert scale: 1–10, with higher scores referring to higher levels of fatigue) were submitted to a RM-ANOVA, with WS factors time (T1–T5) and drug (HAL/PLA). A main effect of time was observed, with fatigue rising across time ($F_{(4,88)} = 6.652$, $p<0.001$, $\eta_p^2 = 0.232$). No main or interaction effect(s) involving drug were observed.

## Appendix 2

### Randomisation groups

For both the social-primary and individual-primary groups, the probability of reward associated with the blue/green stimuli (individual information) and the red stimuli (social information) was governed by different pseudo-randomisation schedules, adapted from *Behrens et al., 2008*. Schedules were counterbalanced between participants to ensure that learning could not be explained in terms of differences in learning between schedules with increased/decreased or early/late occurring, volatility. The individual-primary group (schedules 1,3) were subdivided into two groups, such that half started with predominantly correct social information, and half with predominantly incorrect social information, with the same true for the social-primary group (schedules 2,4). The primary information source was always less volatile overall compared to the secondary information source, irrespective of whether it was social or individual. To give an example, the randomisation schedule for group 1 was the same as that employed by *Behrens et al., 2008*. During the first 60 trials, the individual reward history was stable, with a 75% probability of blue being correct. During the next 60 trials, the reward history was volatile, switching between 80% green correct and 80% blue correct every 20 trials. Meanwhile, during the first 30 trials, social information was stable, with 75% of choices being correct. During the next 40 trials, the social information was volatile, switching between 80% incorrect and 80% correct every 10 trials. During the final 50 trials, social information was once again stable, with 85% of choices being incorrect. Randomisation schedules for groups 2, 3, and 4 were inverted and counterbalanced versions of schedule 1 (*Appendix 2—figure 1*).

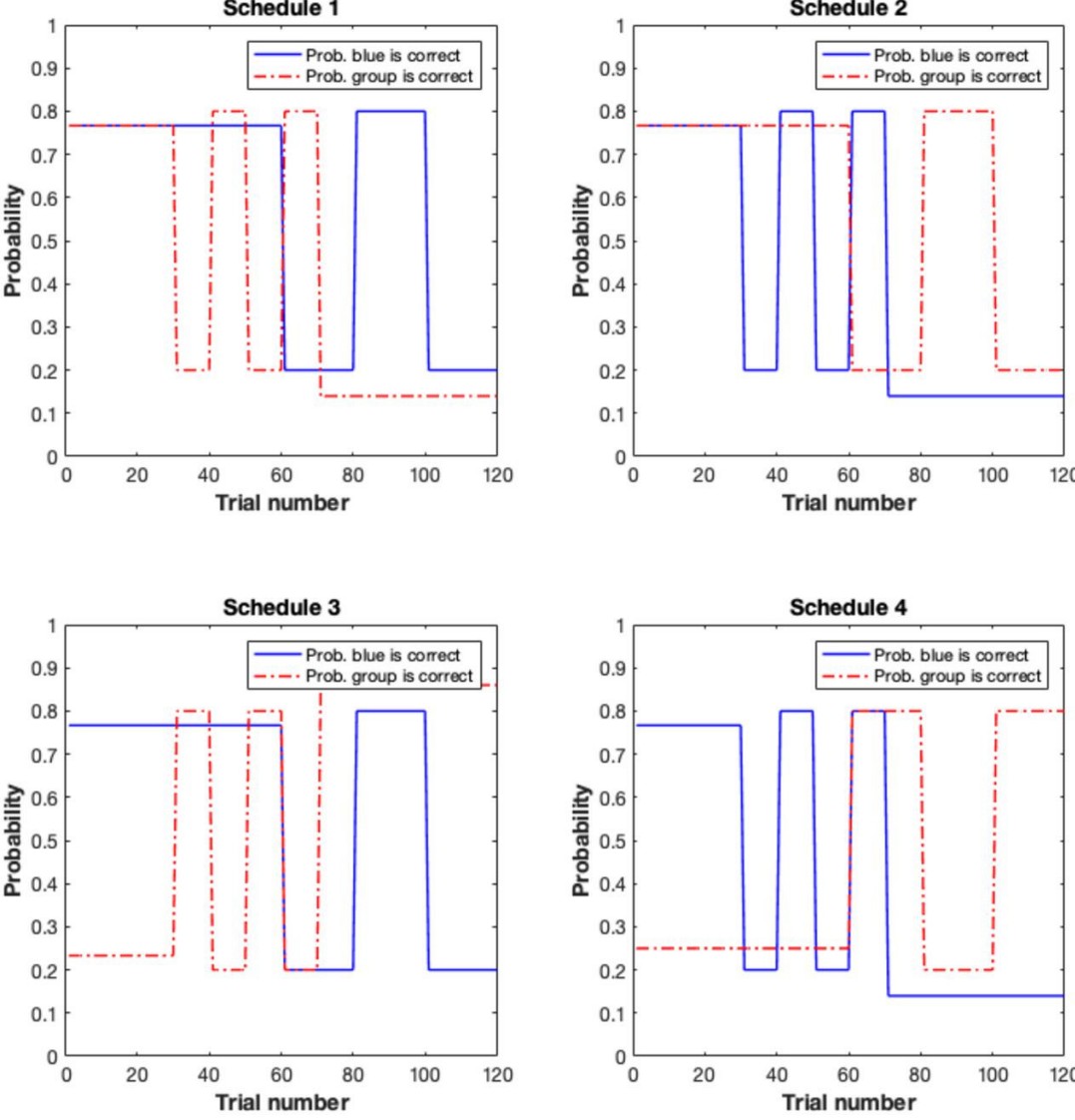

**Appendix 2—figure 1.** Randomisation schedules. The probability of reward varied according to probabilistic schedules, including stable and volatile blocks for both the probability of blue being correct and the probability of the social information indicating the correct answer. Probability schedules were counterbalanced between participants. Solid blue lines show the probability of blue being the correct choice, and dashed red lines show the probability of the social information being correct. Schedules 1–4 are displayed here.

## Supplementary methods

### i. Visual working memory task

Participants completed a visual working memory (VWM) task, adapted from the Sternberg VWM Task (*Sternberg, 1969*), and programmed using MATLAB R2017b. Participants were first presented with instructions followed by practice trials. Upon completion of the practice trials, participants completed 60 experimental trials across five blocks. On each trial, a fixation cross was displayed in the centre of screen (fixation duration varied randomly between 500 and 1000 ms). Then participants were presented with a list of letters (varying between 5 and 9 consonants in length, with letters randomly selected from the alphabet on each trial) for 1000 ms, followed by a blue fixation cross for 3000 ms. Following this, a single test letter was displayed (for a maximum of 4000 ms), requiring participants to determine whether the letter was taken from the previously displayed list. For 50% of trials, the letter had been present on the previous list and on 50% of trials, it had not. Participants responded by pressing 1–3 on the keyboard (1 – yes, 2 – no, 3 – unsure). The total task duration was

approximately 10 min. Responses (accuracy) and response time (time from test letter displayed until participant response) were recorded for each trial.

## ii. Go-No-Go learning

An adapted version of a probabilistic Go/No-Go Task (*Frank and O'Reilly, 2006*) was employed, presented using MATLAB R2017b. In this task, a 'Go' response measures sensitivity to reward, whereas a 'No-Go' response measures sensitivity to punishment. Participants were presented with four different stimuli, each with a probabilistic value of reward (80, 60, 40, and 20%) and instructed to accumulate as many points as possible and to avoid losing points, achieved by selecting or withholding a response to the given stimuli. For example, if selected, stimuli A would result in gaining a point on 80% of trials and losing a point on 20% of trials. Participants were informed that points would be rewarded with monetary compensation; however, due to ethical considerations, all participants were awarded £5 at the end, regardless of task performance. Participants first completed four blocks of a practice stage, where single stimuli were presented (40 trials/block, with each stimulus presented 10 times per block). Reward feedback was provided, allowing learning of the probabilistic value of each stimulus. This was followed by six testing blocks (40 trials/block) displaying either single stimuli (training stimuli) or novel pairs of stimuli on each trial, whereby participants were required to respond based on the *combined* probabilistic value of the pairs. Testing blocks contained positive pairs with a high associated probabilistic reward value, equal pairs (equally probable reward value), and negative pairs, with a high probabilistic value for punishment. Participants could respond via a 'Go' (space bar press) or 'No-Go' (withhold response) response. Feedback was not provided during testing blocks. In all trials, a fixation cross was presented for 250–750 ms, followed by stimuli presentation for 1000 ms and a response period for 250 ms. Task performance was calculated as the difference in 'Go' response for stimuli (novel pairs and single stimuli) with a high probability of reward under HAL and PLA conditions, for each participant separately.

## Appendix 3

### Model fitting

Optimisation of free parameter values was performed as per *Cook et al., 2019* using a quasi-Newton optimisation algorithm specified in TAPAS toolbox – quasinewton_optim_config.m. The function maximised the log-joint posterior density over all parameters given the data and the generative model. **α** values were estimated in logit space (see tapas_logit.m), that is, a logistic sigmoid transformation of native space (tapas_logit(x) = ln(x/(1-x)); x = 1/(1 + exp(-tapas_logit(x)))). An uninformative prior, allowing for individual differences in learning rate, was used for **α**: tapas_logit (0.2, 1), with a variance of 1. Initial values were set at logit (0.5, 1), with a variance of 1. Initial values were allowed to vary, to allow for inter-individual differences in prior preferences for the extent to which individual would conform to the group choice. The prior for **β** was set to log (48), with a variance of 1, and the prior for $\zeta$ was set at 0 with a variance of $10^2$ (logit space), that is, an equal weighting for information derived from primary and secondary learning (0.5). Prior choices were based on previous work (*Cook et al., 2019*). Maximum a posteriori (MAP) estimates for all model parameters were calculated using the HGF toolbox version 3 (https://osf.io/398w4/files/). All codes used were adapted from the open-source software package TAPAS (available at http://www.translationalneuromodeling.org/tapas).

### Model comparison

We based our choice of perceptual model on previous work by *Cook et al., 2019*, wherein a systematic comparison of three alternative models was conducted, to determine which best explained observed choice behaviour. Here, we repeated Cook et al.'s model comparison and added four further extensions of the classic model, thus we compared eight alternative models in total. A formal model comparison was carried out using BMS using the VBA toolbox (*Stephan et al., 2009*).

Data were initially analysed with eight models. All models were variations of the classic RW model. Group-level BMS was used to evaluate which model provided the (relative) best fit to the observed data. The VBA toolbox (*Diaconescu et al., 2014*), specifically random effects BMS (using the VBA_groupBMC_btwConds.m function), was utilised. Random effects group BMS computes an approximation of the model evidence relative to the other models, thats is, the probability of the data *y* given a model *m*, p(y|m), with log model evidence here approximated with F values. The posterior probability that a model has generated the observed data, relative to other models, is estimated, and the exceedance probability, or the likelihood that a given model is more likely than other included models in the set, is estimated. Analysis across both conditions allows us to test the hypothesis that the same model produced observed data under both HAL and PLA conditions.

Model 1 was a classic RW model:

$$V_{(i+1)} = V_i + \alpha \varepsilon_i$$

with $\varepsilon_i = r_i - V_i$, the difference between the actual and the expected reward or prediction error (PE).

Model 2 was an extension of model 1, with separate learning rates ($\alpha$) for learning from primary value and secondary value learning sources:

$$V_{primary(i+1)} = V_{\_primary(i)} + \alpha_{primary} \varepsilon_i$$

$$V_{secondary(i+1)} = V_{secondary(i)} + \alpha_{secondary} \varepsilon_i$$

Model 3 had a single learning rate $\alpha$ for primary/secondary learning, but separate learning rates for volatile and stable blocks:

$$V_{(i+1)} = V_i + \alpha_{\_volatile} \varepsilon_i + \alpha_{\_stable} \varepsilon_i$$

Model 4 had four separate learning rates $\alpha$ for volatile and stable and primary and secondary learning:

$$V_{primary(i+1)} = V_{primary(i)} + \alpha_{primary\_volatile} \varepsilon_i + \alpha_{primary\_stable} \varepsilon_i$$

$$V_{secondary(i+1)} = V_{secondary(i)} + \alpha_{secondary\_volatile}\varepsilon_i + \alpha_{secondary\_stable}\ \varepsilon_i$$

As an exploratory measure, we further extended models 1–4 to include separate learning rates corresponding to learning from rewarded trials and unrewarded trials separately, that is, learning from wins and losses.

Model 5:

$$V_{(i+1)}\ = V_i + \alpha_{\_reward}\ \varepsilon_i + \alpha_{\_unreward}\ \varepsilon_i$$

Model 6:

$$V_{primary(i+1)} = V_{primary(i)} + \alpha_{primary\_reward}\ \varepsilon_i + \alpha_{primary\_unreward}\ \varepsilon_i$$

Model 7:

$$V_{(i+1)}\ = V_i + \alpha_{\_volatile\_reward}\varepsilon_i + \alpha_{\_stable\_reward}\ \varepsilon_i + + \alpha_{\_volatile\_unreward}\ \varepsilon_i + \alpha_{\_stable\_unreward}\ \varepsilon_i$$

Model 8:

$$V_{primary(i+1)}\quad =\quad V_{primary(i)}\quad +\quad \alpha_{primary\_volatile\_reward}\varepsilon_i\quad + \quad \alpha_{primary\_stable\_reward}\ \varepsilon_i\quad + + \alpha_{primary\_volatile\_unreward}\ \varepsilon_i + \alpha_{primary\_stable\_unreward}\ \varepsilon_i$$

$$V_{secondary(i+1)}\quad =\quad V_{secondary(i)}\ V_{(i+1)}\quad +\quad \alpha_{secondary\_volatile\_reward}\varepsilon_i\quad +\quad \alpha_{secondary\_stable\_reward}\ \varepsilon_i\quad + \alpha_{secondary\_volatile\_unreward}\ \varepsilon_i + \alpha_{secondary\_stable\_unreward}\ \varepsilon_i$$

We ran a between-groups model comparison to ensure that the same model could explain the observed data under both PLA and HAL. When comparing all models, model 4 performed best, with an exceedance probability approaching 1. The exceedance probability that the same model (model 4) had produced data under both conditions was equal to 1. For condition 1 (PLA), the posterior probabilities that the observed data had produced the model was equal to 10.329 for model 3 and 12.998 for model 4, with the probability that the data was produced by the winning model p(H1|y) = 0.762. For group 2 (HAL), model 4 had a posterior probability of 15.417 (p(H1|y) = 0.998). For the between-groups assessment, the posterior probability p(H1|y) = 0.999 and the protected exceedance probability (φ) was equal to 0.999.

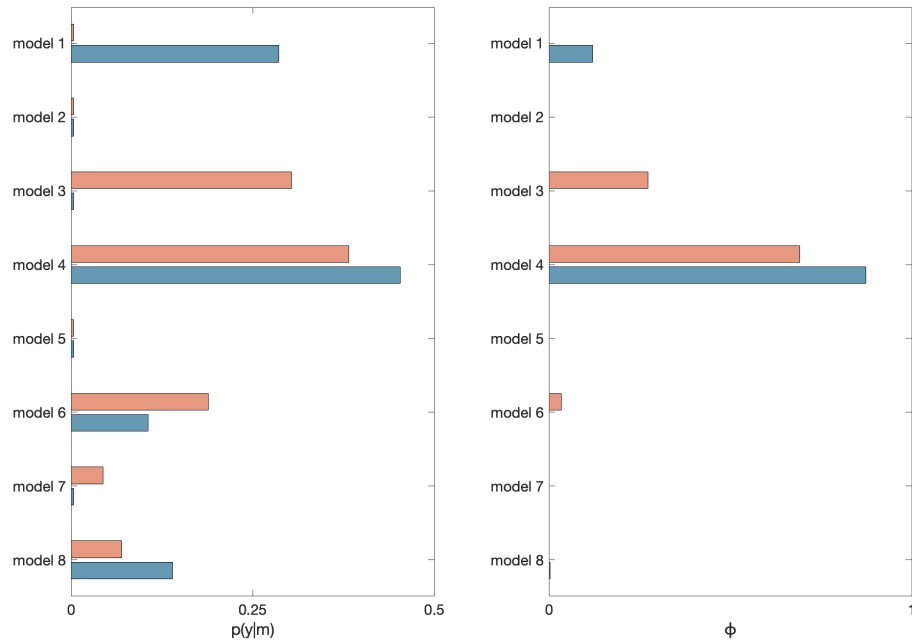

**Appendix 3—figure 1.** Model comparison. Results from random effects Bayesian model selection. Exceedance probability and posterior model probability for models 1–8. p(y|m): posterior model probability; $\phi$: exceedance probability; haloperidol (HAL): blue; placebo (PLA): red.

## Model validation

To demonstrate that the chosen model (model 4) accurately described participant behaviour, we simulated response data for each participant using estimated model parameter values (tapas_simModel.m). Accuracy did not significantly differ between actual and simulated accuracy for PLA (t = –0.866, p=0.394) or HAL conditions (t = –0.280, p=0.781) (*Appendix 3—figure 2A*). Simulated and calculated accuracy was significantly correlated for each participant under both PLA (r = 0.487, p=0.005) and HAL conditions (r = 0.712, p<0.001) (*Appendix 3—figure 2B*).

**A**

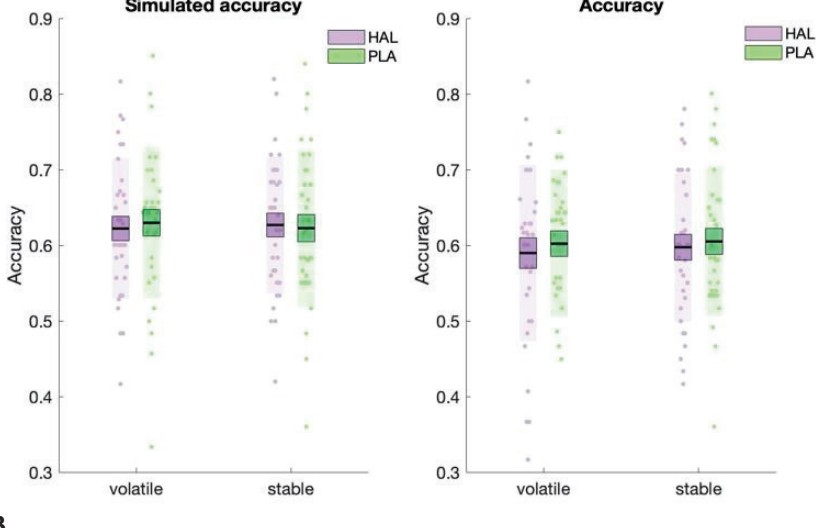

**B**

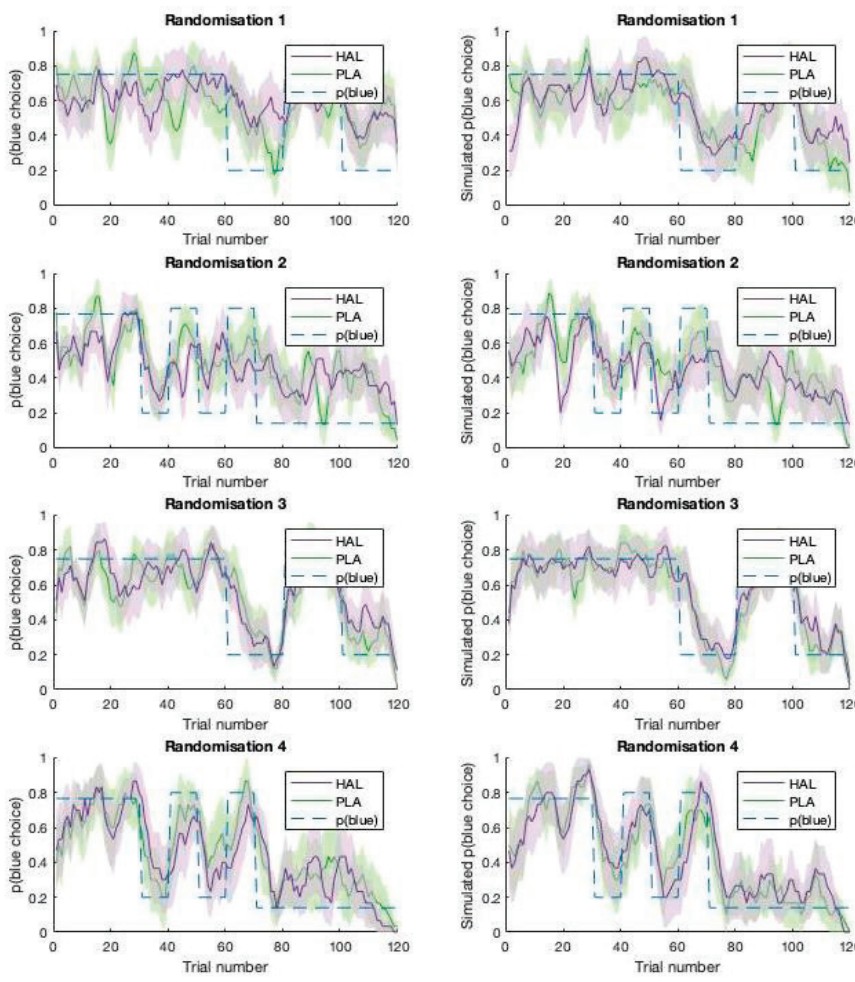

*Appendix 3—figure 2 continued on next page*

*Appendix 3—figure 2 continued*

**Appendix 3—figure 2.** Model validation. (**A**) Model simulations (left) and participant response data (right). Mean accuracy is displayed separately for volatile and stable environmental phases, under haloperidol (HAL; purple) and placebo (PLA; green). Boxes: standard error of the mean; shaded region: standard deviation; individual datapoints are displayed. (**B**) Participant data (left) juxtaposed against model simulations (right). Running average, across five trials of blue choices for probabilistic randomisation schedules 1–4. Shaded region: standard error of the mean.

In addition, to formally test model predictions of choice behaviour, for each participant we calculated the average value that the model estimated for the options chosen by the participant (collapsed across HAL and PLA conditions), and the average value that the model estimated for the options that were not chosen by the participant. If the chosen model was accurately describing participants' choice behaviour, then the average estimated values for chosen options should be significantly higher than for the unchosen options. Indeed, a paired samples *t*-test illustrated that model-derived values for chosen options ($\bar{x}(\sigma_{\bar{x}})$ = 0.607 (0.008)) were significantly greater than the values for unchosen options ($\bar{x}(\sigma_{\bar{x}})$ = 0.393 (0.008); t(30) = 12.558, p<0.001).

To ensure that parameter estimates could be recovered, we simulated response data for each participant, based on estimated model parameters, using the function tapas_simModel.m from the TAPAS toolbox. Model parameters were subsequently estimated from simulated data and averaged over 100 iterations for each participant, separately for HAL and PLA conditions. All recovered parameters correlated significantly with estimated parameters under both HAL ($\alpha_{primary}$: *r* = 0.991, p<0.001, $\alpha_{secondary}$: *r* = 0.961, p<0.001) and PLA ($\alpha_{primary}$: *r* = 0.975, p<0.001, $\alpha_{secondary}$: *r* = 0.984, p<0.001) treatment conditions. A RM-ANOVA on recovered parameters showed the same pattern of results as with estimated parameters, including a significant interaction effect for our main interaction of interest (drug by information source: (F(1,29) = 4.027, p=0.054, $\eta_p^2$ = 0.122)).

## Appendix 4

## Extended statistical analyses

### i. Learning rate analysis (n = 41)

An RM-ANOVA, with (square-root transformed) learning rate ($\alpha$) as the DV and predictors information source, volatility, drug, and group, was carried out on estimates from the mixed model analysis which included all participants who completed at least one study day (N = 41). A significant main effect of information was observed ($F(1,234)$ = 3.944, p=0.048, beta estimate (SE) = 0.019 (0.010); t = 1.986, CI [0–0.04]), with higher mean values for $\alpha_{primary}$ ($\bar{x}(\sigma_{\bar{x}})$ = 0.429 (0.018)) compared with $\alpha_{secondary}$ ($\bar{x}(\sigma_{\bar{x}})$ = 0.391 (0.018)).

A significant volatility by information interaction ($F(1,234)$ = 4.676, p=0.032, beta estimate (SE) = 0.021 (0.010), t = –2.162, CI [0–0.04]) was observed. Post hoc comparisons revealed that, under stable phases, $\alpha_{primary}$ values ($\bar{x}(\sigma_{\bar{x}})$ = 0.461 (0.023)) were significantly greater than $\alpha_{secondary}$ (estimate (SE)$\bar{x}(\sigma_{\bar{x}})$ = 0.381 (0.023); z = 2.933, $p_{holm}$ = 0.007), with no difference between $\alpha$ in volatile phases (z = –0.125, $p_{holm}$ = 0.901). No main effect of group was observed; however, there was a significant information by group interaction ($F(1, 234)$ = 32.471, p<0.001, beta estimate (SE) = 0.05 (0.010); t = 5.700, CI [0.04–0.07]). Post hoc comparisons revealed that, for the individual-primary group, $\alpha_{primary}$ ($\bar{x}(\sigma_{\bar{x}})$ = 0.455 (0.026)) was significantly greater than $\alpha_{secondary}$ ($\bar{x}(\sigma_{\bar{x}})$ = 0.307 (0.026); z = 5.351, $p_{holm}$ < 0.001). For the social-primary group, however, $\alpha_{secondary}$ ($\bar{x}(\sigma_{\bar{x}})$ = 0.475 (0.025)) was significantly greater than $\alpha_{primary}$ ($\bar{x}(\sigma_{\bar{x}})$ = 0.404 (0.025); z = 2.667, $p_{holm}$ = 0.015).

A significant volatility by group interaction was observed ($F (1,234)$ = 4.168, p=0.042, beta estimate (SE) = 0.020 (0.010); t = 2.042, CI [0–0.04]). For the individual-primary group, $\alpha_{volatile}$ ($\bar{x}(\sigma_{\bar{x}})$= 0.351 (0.026)) was (marginally) significantly lower than $\alpha_{stable}$ ($\bar{x}(\sigma_{\bar{x}})$ = 0.411 (0.026), z = –2.192, $p_{holm}$ < 0.057). For the social-primary group, however, $\alpha_{volatile}$ ($\bar{x}(\sigma_{\bar{x}})$= 0.449 (0.025)) and $\alpha_{stable}$ ($\bar{x}(\sigma_{\bar{x}})$ = 0.431 (0.025)) did not significantly differ (z = 0.672, $p_{holm}$ = 0.502). Most importantly, as with the analysis reported in the main text, a significant drug by information interaction was observed ($F (1,234)$ = 3.727, p=0.054, beta estimate (SE) = 0.01 (0.1); t = 1.69, CI [0.00–0.04]). Post hoc comparisons demonstrated that, under PLA, there was a significant difference between $\alpha_{primary}$ ($\bar{x}(\sigma_{\bar{x}})$ = 0.451 (0.023)) and $\alpha_{secondary}$ ($\bar{x}(\sigma_{\bar{x}})$ = 0.375 (0.023); z = 2.727, $p_{holm}$ = 0.026, uncorrected p=0.006). This difference was nullified under HAL ($\alpha_{primary}$ $\bar{x}(\sigma_{\bar{x}})$ = 0.408 (0.023) and $\alpha_{secondary}$ $\bar{x}(\sigma_{\bar{x}})$ = 0.407 (0.023), z = 0.040, $p_{holm}$ = 0.968, uncorrected p=0.968). There was no significant group × information source × drug interaction ($F(1,234)$ = 0.029, p=0.866, beta estimate (SE) = –0.002 (0.010), t = –0.169, CI [-0.02–0.02]).

### ii. Accuracy

An analysis of accuracy was conducted in participants who had completed both study days (n = 31) to explore whether there was any systematic variation as a function of randomisation schedule and across drug and PLA conditions and volatile and stable phases. An RM-ANOVA, with WS factors drug (HAL, PLA) and volatility (stable, volatile), and between-subjects factor group (social-primary, individual-primary) and randomisation schedule (1–4), demonstrated no difference in accuracy between HAL ($\bar{x}(\sigma_{\bar{x}})$ = 0.601 (0.011)) and PLA ($\bar{x}(\sigma_{\bar{x}})$ = 0.614 (0.011)); $F(1,27)$ = 1.161, p=0.291, $\eta_p^2$ = 0.041). However, a significant main effect of schedule was observed ($F(3,27)$ = 3.004, p=0.048, $\eta_p^2$ = 0.250), with the lowest accuracy observed for schedule 1 ($\bar{x}(\sigma_{\bar{x}})$ = 0.558 (0.019)). Although accuracy for schedule 1 was lower than for schedule 2 ($\bar{x}(\sigma_{\bar{x}})$ = 0.619 (0.018); t (27) = –2.358, $p_{holm}$ = 0.129), schedule 3 ($\bar{x}(\sigma_{\bar{x}})$ = 0.614 (0.018); t(27) = (–2.162), $p_{holm}$ = 0.159), and schedule 4 ($\bar{x}(\sigma_{\bar{x}})$ = 0.637 (0.020); t(27) = –2.748, $p_{holm}$ = 0.063); these differences were no longer significant after correction for multiple comparisons. Mean accuracy for schedules 2–4 did not significantly differ from each other (all p-values = 1.000). In addition, there was a significant interaction effect between schedule and volatility ($F(3,27)$ = 7.527, p<0.001, $\eta_p^2$ = 0.455). For all schedules except for schedule 3, there was no significant difference in accuracy between volatile and stable phases (all p>0.05). However, for schedule 3, accuracy was significantly higher for volatile ($\bar{x}(\sigma_{\bar{x}})$ = 0.675 (0.022)) over stable phases ($\bar{x}(\sigma_{\bar{x}})$ = 0.533 (0.022); t(27) = (3.656), $p_{holm}$ = 0.027). Accuracy was significantly higher for the social-primary group ($\bar{x}(\sigma_{\bar{x}})$ = 0.629 (0.013)) compared with the individual-primary group ($\bar{x}(\sigma_{\bar{x}})$ = 0.586 (0.013), $F(1,29)$ = 5.196, p=0.030, $\eta_p^2$ = 0.152), and no other main effects or interactions were observed (all p>0.05).

### iii. Relationship between accuracy scores and parameters from model-based analyses

A backward regression with PLA accuracy as the dependent variable, and $\alpha_{primary}$ and $\alpha_{secondary}$ (collapsed across volatile and stable phases), initial values $V_{primary(i)}$ and $V_{secondary(i)}$, $\beta$ and $\zeta$ as predictors, was carried out. PLA accuracy was marginally significantly predicted by a model with $\alpha_{secondary}$ as a single predictor ($R$ = 0.347, F(1,29) = 3.981, p=0.055). Under HAL, a backward regression with HAL accuracy as the dependent variable, and $\alpha_{primary}$, $\alpha_{secondary}$, $V_{primary(i)}$, $V_{secondary(i)}$, $\beta$ and $\zeta$ as predictors, revealed that HAL accuracy was significantly predicted by the full model. Within the model, $\alpha_{primary}$ was the only significant predictor (*Appendix 4—table 1*). Removing predictors did not significantly improve the fit of the model ($R^2$ change < 0.001, F change (1,25) = –0.064, p=1.000).

**Appendix 4—table 1.** Coefficients from regression model with haloperidol (HAL) accuracy as the dependent variable.

|  | β | β (SEM) | Standardised β | t | p-Value |
|---|---|---|---|---|---|
| *Constant* | 0.431 | 0.089 |  | 4.840 | <0.001 |
| $\alpha_{primary}$ | 0.195 | 0.077 | 0.431 | 2.532 | 0.018* |
| $\alpha_{secondary}$ | 0.076 | 0.119 | 0.127 | 0.642 | 0.527 |
| $V_{primary(i)}$ | 0.121 | 0.090 | 0.230 | 1.342 | 0.192 |
| $V_{secondary(i)}$ | 0.033 | 0.131 | 0.050 | 0.249 | 0.806 |
| β | 0.002 | 0.001 | 0.329 | 1.698 | 0.102 |
| ζ | 0.045 | 0.043 | 0.189 | 1.066 | 0.297 |

* indicates statistical significance.

### iv. Go-No-Go control task

To further investigate the neurochemical mechanisms underlying the observed decrease in $\alpha_{primary}$ under HAL, we measured performance on a probabilistic Go-No-go control task adapted from *Frank and O'Reilly, 2006*. Previous research (using a similar low, acute dose of HAL) resulted in enhancement of learning from positive reinforcement, indexed by an increase in learning from positive feedback (*Frank and O'Reilly, 2006*), suggested to be mediated via presynaptic antagonistic effects on phasic dopamine (DA) signalling. As an exploratory measure, participants were stratified into two subgroups based on performance during this task; those with a higher change in 'Go' performance for high reward trials under HAL, and those with a lower change in 'Go' performance under HAL, relative to PLA. For the participants who demonstrated increased 'Go' performance under HAL (n = 12), a significant drug by information effect was observed on the main behavioural task (F(1,10) = 4.773, p=0.054, $\eta_p^2$ = 0.323). However, this effect was not observed in participants with reduced 'Go' performance under HAL (n = 19; F(1,17) = 2.001, p=0.175, $\eta_p^2$ = 0.105). Thus, suggesting that the observed effect of HAL on learning rate for primary information was driven by a subgroup of participants who exhibited increased 'Go' performance under HAL (relative to PLA). Given that such effects on Go performance have been linked to presynaptic antagonistic effects on phasic DA signalling (*Frank and O'Reilly, 2006*), these results suggest that the effects we observed on $\alpha_{primary}$ are likely mediated by effects of HAL on phasic DA signalling.

While an increase in Go performance suggests presynaptic effects of HAL on phasic dopamine release, the effects of HAL are also mediated via antagonism of heteroreceptors on non-dopaminergic neurons (*Frank and O'Reilly, 2006*), resulting in a reduction in tonic dopamine signalling. These tonic effects are commonly indexed by a slowing of response (*Grace, 2002*; *Niv et al., 2007*). Indeed, HAL had a significant effect on (log) reaction time (RT), with higher RTs observed under HAL ($\bar{x}(\sigma_{\bar{x}})$ = 1.580 (0.147) s) when compared with PLA ($\bar{x}(\sigma_{\bar{x}})$ = 1.242 (0.150), p=0.002, $\eta^2$ = 0.292). We therefore investigated whether there was a relationship between $\Delta RT$ and $\Delta \alpha$ under HAL. A median split ($\Delta RT$) resulted in two subgroups of participants. Separate RM-ANOVAs, with (square root) learning rate estimates ($\alpha$) as the dependent variable, and information, volatility, and task group as the predictor variables, were carried out for each subgroup. For the subgroup of participants who showed the greatest increase in RT (slowing of response) under HAL (n = 15), the drug by information interaction no longer reached significance (F(1,13) = 0.106, p=0.750, $\eta_p^2$ = 0.008). The

opposite pattern of results was observed for the subgroup of participants (n = 16) with a $\Delta RT$ below the median change (a reduced slowing of response under HAL): here, a significant drug by information interaction effect was observed (F(1,14) = 10.846, p=0.005, $\eta_p^2$ = 0.437). Results show that, for the subgroup of participants who showed the greatest slowing of response ($\Delta RT$), HAL did not significantly affect learning rates. Given that response slowing has been linked to tonic dopamine, this pattern of results further reinforces the idea that our observed effects on $\alpha_{primary}$ are likely mediated by effects of HAL on phasic, not tonic, DA.

### v. Effect of randomisation schedule and drug day on model parameters

Randomisation schedule (1–4) and drug day (i.e., HAL administered on testing day 1 or 2) were included as predictor variables in all analyses (with both n = 31 and n = 41 samples), with no main/ interaction effect(s) observed (all $F < 1$, all p>0.05). Additionally, testing session was used to check for the presence of practice effects. Testing session (session 1 or 2) was included as a predictor variable in all analysis, with no main/interaction effect(s) observed (all $F < 1$, all p>0.05).

### vi. Effects of baseline VWM on model parameters

As there is evidence to suggest that the effects of dopamine manipulation are dependent on baseline DA synthesis, with working memory capacity shown to predict dopamine synthesis in healthy adults (***Cools et al., 2008***), we stratified participants into high and low VWM groups, based on mean baseline (under PLA) accuracy scores on a VWM task (***Sternberg, 1969***). VWM (high/low) was included as a predictor in a mixed model analysis (n = 31). A type III RM-ANOVA conducted on model estimates revealed a significant interaction between VWM and information type (F(1,189) = 5.932, p=0.016, beta estimate (SE) = 0.026 (0.010), t = 2.436, CI [0.00–0.05]) with planned contrasts revealing that, for low VWM participants, $\alpha_{secondary}$ values ($\bar{x}(\sigma_{\bar{x}})$ = 0.364 (0.031)) were significantly lower than $\alpha_{primary}$ values ($\bar{x}(\sigma_{\bar{x}})$ = 0.447 (0.031); z(30) = 2.820, $p_{holm}$ = 0.010). There was no significant difference between $\alpha_{primary}$ and $\alpha_{secondary}$ for high VWM participants (z(30) = –0.641, $p_{holm}$ = 0.522). No other main or interaction effects of VWM on $\alpha$ values were observed (all $F < 0.01$, all p>0.05). Additionally, the pattern of results was unchanged from the previous analysis excluding VWM, with the drug by information interaction effect remaining significant (F(1,189) = 3.967, p=0.048, beta estimate (SE) = 0.021 (0.010), t = 1.992, CI [0.00–0.04]). Finally, while including baseline VWM as continuous predictor variable in a RM-ANOVA, no main or interaction effect(s) of VWM on $\alpha$ values were observed. Additionally, neither gender, age, nor BMI interacted with any outcome variables (all $F < 0.01$, all p>0.05). Results suggest that the observed decrease in $\alpha_{primary}$ under HAL is not related to variation in working memory capacity.

## Appendix 5

### Instruction scripts

Individual-primary group

Welcome. You have a choice: either choose the blue shape or the green shape. One shape is correct – guessing which one it is will give you points. To help you to choose, one of the shapes is filled with red. This indicates the most popular choice selected by a group of four people who previously played this task. When the question mark appears, try picking a shape by pressing the left or right keyboard buttons. [Participant responds]

Feedback: After you make a choice, a tick or cross will appear in the middle. This tells you if the group of previous players were correct or incorrect.

Here they think the blue shape (filled with red) will be correct. Try picking a shape now. [Participant responds]

Blue is correct! This means that this time the others got it right.

Things happen in phases in this game. The game could be in a phase where the blue shape is more likely to be correct. Have another go. [Participant responds]

And blue again! It certainly looks as though you are in a blue phase but make sure you pay attention to what the right answers are because the phase that you are in can change at any time. Here's a tip – ignore which side of the screen the shapes are on – it's the colour that is important! [Participant responds]

The others got it right again. It looks like, right now, you could be in a phase where the group's information is useful. Perhaps these are trials from the end of their experiment, when they had developed a pretty good idea of what was going on. Be careful though because we have mixed up the order of the other people's trials so that their choices will also follow phases. Try again. Perhaps the other shape is right this time? [Participant responds]

Green! This time the green shape was right! The chance of each shape being right or wrong will change as you play, so pay attention! The group were incorrect this time. Remember that sometimes you will see less useful information from the group – for example from the beginning of their experiment where they didn't have a very good idea of what was going on. Have another go … [Participant responds]

This time the green shape was right! The chance of each shape being right or wrong will change as you play, so pay attention. The group were correct too. It looks like, right now, you could be in a phase where the group's information is useful. Try to be as accurate as possible. Getting it right, gives you points. Get enough points and you could earn a silver or even a gold prize! Have another go … [Participant responds]

Things happen in phases in this game. Remember, the tick or cross in the middle tells you if the group were correct or incorrect. That means that the shape with the red box was the correct choice. Have another go … [Participant responds]

### Social-primary group

Welcome. You have a choice between going with, or against advice from a group. Below you can see a blue and green frame, one frame is filled with a red box: this indicates the most popular choice selected by a group of four people who previously played this task. One frame is correct. You can pick the same frame as the group have picked or choose to go against the group's advice. When the question mark appears, make your selection by pressing the left or right keyboard buttons. [Participant responds]

Feedback: After you make a choice, a tick or cross will appear in the middle. This tells you if the group of previous players were correct or incorrect.

This time they were correct! This means that the frame filled with the red square was the correct frame.

Here they think the blue frame (filled with red) will be correct. Try picking a frame now. [Participant responds]

The group were correct! This means that this time the others got it right and picked the correct colour.

Things happen in phases in this game. The game could be in a phase where the group are more likely to be correct. Have another go. [Participant responds]

The group were correct again! The blue frame was right again. It certainly looks as though you are in a phase where the group are correct but make sure you pay attention to the feedback because the phase that you are in can change at any time. Blue and green can also go through phases: it looks like you might be in a phase where the blue frame is more likely to be correct. Try again. [Participant responds]

The others got it right again. It looks like, right now, you could be in a phase where the group's information is pretty useful. Perhaps these are trials from the end of their experiment, when they had developed a pretty good idea of what was going on. Be careful though because we have mixed up the order of the other people's trials so that their choices will follow phases. Try again. [Participant responds]

The group were incorrect this time. This time the green frame was correct. The chance of each frame being right or wrong will change as you play, so pay attention! Remember that sometimes you will see less useful information from the group – for example from the beginning of their experiment where they didn't have a very good idea of what was going on. Have another go … [Participant responds]

The group were correct this time. The chance of each frame being right or wrong will change as you play, so pay attention. Try to be as accurate as possible. Getting it right, gives you points. Get enough points and you could earn a silver or even a gold prize! Have another go … [Participant responds]

Things happen in phases in this game. Remember, the tick or cross in the middle tells you if the group were correct or incorrect. That means that the frame filled with the red was the correct choice. Have another go … [Participant responds]

## Feedback questionnaire

Participants competed a short feedback questionnaire after the behavioural task. 100% of participants said that they understood the task instructions and what they were supposed to do. Participants were then asked to rate on a 5-point Likert scale how often they (1) used the group's suggestions (red shape) to help make their decision, comprising the social rating score, and (2) if they paid attention to the colour of the shape (blue/green) that was correct when making their decision (the individual rating score). Social and individual ratings were submitted to separate one-sample $t$-tests to ensure that participants in both the individual-primary and social-primary groups were paying attention to both sources of information. Both social ($t(42) = 30.765$, $p<0.001$) and individual ratings ($t(42) = 29.565$, $p<0.001$) were significantly greater than zero.

