## [Editor Report]

This work has important implications on how we view and understand social and individual learning with respect to dopamine processing in the human brain. This study, supported by a well-controlled experimental design, clear hypothesis testing, and rigorous model-based analyses, revealed that the dopamine system is involved in learning from a primary source as opposed to a secondary source, irrespective of social or non-social individual learning. This work encourages new investigations into testing when and how different neuromodulator systems may converge or diverge in guiding social versus non-social learning.

---

## [Decision Letter]

**Decision letter after peer review:**

Thank you for submitting your article "Dopaminergic challenge dissociates learning from primary versus secondary sources of information" for consideration by *eLife*. Your article has been reviewed by 3 peer reviewers, including Steve Chang as the Reviewing Editor and Reviewer #1, and the evaluation has been overseen by Floris de Lange as the Senior Editor.

Please note that the three reviewers have agreed on the Essential Revision items, which are shown below. Moreover, there are additional comments that could be incorporated in your revision but not what required as we emphasize the revision addressing the Essential Revision Items.

Essential revisions

1) The domain specificity or lack thereof depends on which level of analysis one focuses on. For example, in one specific region, dopamine-dependent social learning may be computed in a socially specific manner even though the behavioral consequences appear to be domain-general. Furthermore, there is an argument to be made with regards to Marr's three levels – computational, algorithmic, and implementational – when one considers if a system (brain regions, circuits, neurodmodulator, etc) exhibits social specificity. For example, social specificity may reside at the algorithmic level but not at the implementational level (or vice versa). The authors may want to discuss some limitation in concluding the domain-general interpretation arising from systemically manipulating haloperidol and only examining the behavioral output.

2) Haloperiodol made participants more optimal by reducing the difference between learning rate-α and the optimal-α (Figure 4). The authors cite research supporting that haloperidol can reduce learning rate by attenuating prediction error signaling. This seems like an impairment in learning, yet haloperidol made the primary-source learning to become more optimal. I suggest the authors to more fully discuss how to understand these results in light of the results up that point in their discussion. It seems like what the authors have provided in the discussion on this issue could be further expanded to make this part of the finding more understandable with respect to the entire results of the paper.

3) There are quite a number of analyses here--ANOVAs, mixed models, Bayes Factors, many of which involve parameters themselves fitted to data--and there is not much to harmonize these approaches. More importantly, with so many analyses (and at least a handful of parameters to test for each), there is real danger of a garden-of-forking paths problem in these results. Not only because learning rates can be unstable to estimate with small numbers of trials, but because learning rates do not appear (from the pre-registration) to have been part of the initial set of dependent behavioral variables to be analyzed. Conversely, effects on the social/non-social balance zeta do not appear in the main text, though these surely were also considered. To this end, I very much appreciated the results in Supp. Figure 3, which ate least showed that the fitted models reproduced the data reasonably well, but any other efforts the authors could make to robustify their central result would be welcome.

4) While the manuscript is very tightly written, I would have appreciated a little more signposting for the secondary analyses. The main effect in Figure 3 seems clear enough, but there are so many tests and secondary analyses (e.g., ll. 267-292) that I found I got bogged down around the analyses leading to Figure 4. Even a few extra comments to reinforce the big picture would be appreciated.

5) Optimal learner analysis – I did not find the details about the analysis in the methods/appendices – maybe I missed them? Could you highlight them?

6) Optimal learner analysis – this seems like a very interesting analysis step, why did you not use it in the HAL condition to characterize its effects in this respect?

7) Learning rate estimates minus optimal learning rates – while optimal learning rates are around 0.11, and estimated learning rates are around 0.4, the difference between them is reported to be around 0 (Figure 4). I am probably missing something here but I am not sure what.

8) There are some discrepancies in the description of results of the optimal β learner results. For example, it indicates that the average β for primary is 0.872 (line 232), but the average of social-primary is 0.833 and the average of ind-primary is 0.477, whose average is ~0.6 – can you explain this? Also, you indicate that there is no interaction effect, but the differences observed in the graph (Figure 2) and the reported effects suggest that there may be a significant effect (social primary-secondary > ind primary-secondary)?

9) Learning rates effects – it is not clear whether volatility had any effect, and how it is incorporated in analysis – for example in Figure 3 and the text (line 280). You state α_primary, but not whether this is during volatile or stable periods. Did you average the learning rates between periods? Used random effect for these observations? This is also the case for the optimal learning rate analysis.

Additional Comments

1) I found the lack of a significant main effect of drug on accuracy very surprising. Other behavioral differences such as response times between the social-primary and individual-primary groups due to haloperidol relative to placebo could have potentially affected choices in different ways. Were there any differences in accuracy with respect to response times due to haloperidol vs. placebo?

2) There were social-primary group and individual-primary group. I am curious if there were fluctuations (within the group) on how much people relied upon social versus individual sources of information across different trials, and if these fluctuations could have differently influenced the effects of haloperidol.

3) Using the TAPAS toolbox, which has explicit volatility estimation model similar to the one used by Behrnes et al., 2008, raises the question whether using such model which directly evaluates volatility may be more useful in analysis of learning pattern here. Put in other words, would you lose anything if volatility was not changed during the task – just a stable period of 0.7 probability of accurate advice/blue, for example?

4) I find the result that primary learning from social info is better (more similar to optimal learning – β value analysis and Figure 2) than primary learning from non-social information very interesting. It indicates that, when cast as primary task, people are very attuned to social information and can make a good use of it. This is something we now see in works where people learn about people's traits, such as honesty (Bellucci 2019). I wonder if this result and the prospect of recasting social tasks can be given some more attention.

---

## [Author Response]

Essential revision:1) The domain specificity or lack thereof depends on which level of analysis one focuses on. For example, in one specific region, dopamine-dependent social learning may be computed in a socially specific manner even though the behavioral consequences appear to be domain-general. Furthermore, there is an argument to be made with regards to Marr's three levels – computational, algorithmic, and implementational – when one considers if a system (brain regions, circuits, neurodmodulator, etc) exhibits social specificity. For example, social specificity may reside at the algorithmic level but not at the implementational level (or vice versa). The authors may want to discuss some limitation in concluding the domain-general interpretation arising from systemically manipulating haloperidol and only examining the behavioral output.

We thank the Reviewer for highlighting this relevant debate. Indeed, recent work by the Reviewing Editor and colleagues – now cited on lines 465-467 of the manuscript – has drawn attention to the relevance of Marr’s levels of analyses to debates about social-specificity. Specifically, it has been argued that whilst there is a growing body of evidence suggesting that the same algorithms are used for social and individual learning, nevertheless, dissociations may exist in terms of the underlying cells or circuits. To phrase this in the terminology of Marr’s Levels of Analysis, there may be dissociations between social and individual learning at the implementation but not the algorithmic level (Lockwood et al., 2020).

Our study concerns the impact of disrupting the implementation level on the algorithmic level. That is, we fitted the same algorithm to the social and individual conditions (i.e., based on existing literature (e.g., Behrens et al., 2008; Diaconescu et al., 2014), we assumed a priori that the same algorithm would be appropriate for both social and individual learning) and then we manipulated the implementation level by disrupting dopamine signalling using haloperidol. We concluded that our manipulation had comparable effects at the algorithmic level; that is, manipulating dopamine signalling had comparable effects on learning rates related to social and individual learning. It is, however, important to recognise that the implementation level is complex (i.e., it comprises many neurons, neural circuits, various neurotransmitters etc.) and our study tests only one of many possible components. Although we focused on dopamine because of its prominence in the learning literature, it is, however, possible that other forms of disruption at the implementation level (e.g., pharmacological manipulation of the *serotonin* system) could dissociate social from individual learning. We highlight this possibility in the following text:

… “it is possible that social and individual learning share common *dopaminergic* mechanisms when they are the primary source, but differentially recruit other neurochemical systems. For instance, some have argued that social learning may heavily rely upon serotonergic mechanisms (Crişan et al., 2009; Frey and McCabe, 2020; Roberts et al., 2020).” (p. 19, lines 482-485)

It is also possible that, we may have seen differences in the *location* of neural activity if we had combined our pharmacological manipulation with neuroimaging. Our current results would predict that haloperidol comparably affects the BOLD signal associated with social and individual prediction errors (when they are the primary source of learning), but it may be that the effects are localised to, for example, the temporoparietal junction and striatum respectively (in line with Behrens et al., 2008). Nevertheless, whilst such location-based differences are *possible*, we argue that they are not *probable* for two reasons*.* First, it is not easy to conceive of how location-based differences in the absence of behavioural differences, would arise (either ontogenetically or phylogenetically). If there is no outward behavioural signal that differentiates people who have two neural pathways (for social and individual learning) from those who have one single pathway, then how would individuals with two pathways be “selected for” (either via natural selection or caregiver reinforcement). Second, given different distributions of dopamine neurons, receptors and reuptake mechanisms throughout the brain, differences in location are relatively likely to result in differences in the magnitude of the effect of haloperidol on learning rates (or other parameters in our computational model). We thank the Reviewers and Editors for encouraging us to reflect on this more fully and have now made the following changes to the manuscript:

“It is possible that although social and individual learning are affected by dopaminergic modulation – when they are the primary source -, there are differences in the *location* of neural activity that could be revealed by neuroimaging. For instance, although social and individual learning are both associated with activity within the striatum (Burke et al., 2010; Cooper et al., 2012), social-specific activation patterns have been observed in other brain regions, including the temporoparietal junction (Behrens et al., 2008; Lindström et al., 2018) and the gyrus of the anterior cingulate cortex (Behrens et al., 2008; Hill et al., 2016; Zhang and Gläscher, 2020). Consequently, it is possible that haloperidol has comparable effects on social and individual learning but that these effects (seen at an “algorithmic level of analysis”, Lockwood et al., 2020) are associated with activity in different brain regions (i.e., dissociations at an “implementation level of analysis”; Lockwood et al., 2020). For example, haloperidol may comparably affect the BOLD signal associated with social and individual prediction errors, but the effect may be localised to dissociable neural pathways. Such a location-based dissociation requires further empirical investigation as well as further consideration of the possible functional significance of such location-based differences, if they are indeed present when primary versus secondary status is accounted for. Nevertheless, whilst such location-based differences are *possible*, we argue that they are not *probable* since, given different distributions of dopamine neurons, receptors and reuptake mechanisms throughout the brain (Grace, 2002; Korn et al., 2021; Matsumoto et al., 2003; Sulzer et al., 2016), differences in location are relatively likely to result in differences in the magnitude of the effect of haloperidol (Wächtler et al., 2020; Yael et al., 2013).” (p. 18, lines 457-477)

2) Haloperiodol made participants more optimal by reducing the difference between learning rate-α and the optimal-α (Figure 4). The authors cite research supporting that haloperidol can reduce learning rate by attenuating prediction error signaling. This seems like an impairment in learning, yet haloperidol made the primary-source learning to become more optimal. I suggest the authors to more fully discuss how to understand these results in light of the results up that point in their discussion. It seems like what the authors have provided in the discussion on this issue could be further expanded to make this part of the finding more understandable with respect to the entire results of the paper.

We agree with the Reviewer that it was somewhat unexpected that, under haloperidol, the observed decrease in learning rates resulted in a closer-to-optimum performance. We thank the Reviewers for inviting us to reflect further on this and have now added the following section to the Discussion:

“Notably, our results reveal a clear dissociation between learning from primary and secondary sources. For learning from primary sources haloperidol made learning rates more optimal, haloperidol did not have this effect on learning rates for secondary learning. Interestingly, a combined optimality analysis and regression model suggested that, under placebo, learning rates for learning from the primary source were “too high” and fell outside of the optimal range (for this specific task). Consequently, under placebo, variance in accuracy was primarily explained by learning rates for learning from the secondary source. However, haloperidol reduced learning rates for learning from the primary source, bringing them within the optimal range. Thus, under haloperidol, accuracy was driven by learning rates for learning from both the primary and secondary sources. An open question concerns whether haloperidol truly *optimises*, or simply *reduces* learning rate. Since the current paradigm was not designed to test this hypothesis a reduction in learning rate herein also corresponds to an optimisation of learning rate. To dissociate the two, one would need a paradigm that generates sufficient numbers of participants with learning rates (in the placebo condition) that are *sub-optimally low* such that one can observe whether, in these critical test cases, haloperidol *increases* (i.e., optimises) learning rate. (Lines 490-504)

An intriguing question concerns the synaptic mechanisms by which haloperidol affects learning rate. Non-human animal studies, have shown that phasic signalling of dopaminergic neurons in the mesolimbic pathway encodes reward prediction error signals (Schultz, 2007; Schultz et al., 1997). Since haloperidol has high affinity for D2 receptors (Grace, 2002), which are densely distributed in the mesolimbic pathway (Camps et al., 1989; Lidow et al., 1991), dopamine antagonists including haloperidol can affect phasic dopamine signals (Frank and O’Reilly, 2006) – either via binding at postsynaptic D2 receptors (which blocks the effects of phasic dopamine bursts), or via presynaptic autoreceptors (which has downstream effects on the release and reuptake of dopamine and thus modulates bursting itself) (Benoit-Marand et al., 2001; Ford, 2014; Schmitz et al., 2003). That is, haloperidol may affect learning rate via blockade of the postsynaptic D2 receptors, which may mute the effects of phasic dopamine signalling (either directly or via reduction in the background tonic rate of activity which, in turn, reduces the amplitude of phasic responses (Belujon and Grace, 2015; Grace, 2016)), thus reducing the weight of prediction error signals on value updating (i.e., reducing the learning rate). Indeed a number of studies have shown that haloperidol can attenuate prediction error-related signals (Diederen et al., 2017; Haarsma et al., 2018; Menon et al., 2007; Pessiglione et al., 2006). For example, in the context of individual learning, Pessiglione et al., (2006) demonstrated that haloperidol attenuated prediction error signals in the striatum, indexed via changes in blood oxygen levels (BOLD). In addition to effects on postsynaptic D2 receptors, haloperidol may modulate prediction errors via its effects on presynaptic autoreceptors. Autoreceptor binding is suggested to increase phasic bursting (Dugast et al., 1997; Frank and O’Reilly, 2006; Garris et al., 2003; Pehek, 1999) thus enhancing the phasic signal that is indicative of positive prediction errors. A combination of pre- and post-synaptic effects could feasibly result in more optimal learning rates wherein dopamine signalling is muted via postsynaptic blockade thus muting (tonic background) “noise” (and signal) but where the phasic “signal” is enhanced via presynaptic effects, potentially resulting in an overall increased signal-to-noise ratio which may translate into more optimal learning rates.” (p. 20, lines 506-530)

3) There are quite a number of analyses here--ANOVAs, mixed models, Bayes Factors, many of which involve parameters themselves fitted to data--and there is not much to harmonize these approaches. More importantly, with so many analyses (and at least a handful of parameters to test for each), there is real danger of a garden-of-forking paths problem in these results. Not only because learning rates can be unstable to estimate with small numbers of trials, but because learning rates do not appear (from the pre-registration) to have been part of the initial set of dependent behavioral variables to be analyzed. Conversely, effects on the social/non-social balance zeta do not appear in the main text, though these surely were also considered. To this end, I very much appreciated the results in Supp. Figure 3, which ate least showed that the fitted models reproduced the data reasonably well, but any other efforts the authors could make to robustify their central result would be welcome.

We thank the Reviewers for highlighting this. To harmonise the results, we have now added an additional paragraph at the beginning of the Results section (pg. 8-9) where we explain our overall analysis approach and provide a roadmap to help the reader navigate the results:

“We used the following strategy to analyse our data. First, we sought to validate our manipulation by testing (under PLA) whether participants in both the individual-primary and social-primary groups learned in a more optimal fashion from the primary, versus secondary, source of information. Next, we tested our primary hypothesis that both social and individual learning would be modulated by haloperidol when they are the *primary* source of learning, but not when they comprise the *secondary* source. To do so we estimated learning rates for primary and secondary sources of information, for each group (social-primary, individual-primary), under HAL and PLA, by fitting an adapted Rescorla-Wagner learning model to choice data. To ascertain that our model accurately described choices we used simulations and parameter recovery. We used random-effects Bayesian model selection to compare our model with alternative models. These analyses provided confidence that our model accurately described participants’ behaviour. After testing our primary hypothesis, we explored the relationship between parameters from our computational model and performance. To accomplish this, we first used an optimal learner model, with the same architecture and priors as our adapted Rescorla-Wagner model, to assess the extent to which haloperidol made participants’ learning rates more (or less) optimal. Finally, we regressed estimated model parameters against accuracy to gain insight into the extent to which variation in these parameters (and the effect of the drug thereupon) contributed to correct responses on the task.” (p. 8-9, lines 222-238)

The Reviewer notes that … “learning rates do not appear (from the pre-registration) to have been part of the initial set of dependent behavioral variables to be analyzed” …. Although our primary pre-registered analysis focused on win-stay, lose-shift (WSLS) behaviour, which we intended to use as a proxy measure of learning rate, our pre-registration document also stated that learning rates would be explored (see Analysis Plan – Exploratory analysis). When we analysed the data according to a WSLS framework, we discovered that whilst the pattern of results was the same (i.e., the same as the learning rate analysis detailed in the current manuscript), the interaction between drug and information became stronger if we calculated the influence on choices of winning/losing from more historically distant trials (i.e., not simply looking one trial back as one would do for a classic WSLS analysis; e.g. the drug and information type interaction relating to the influence on choices of outcomes from *two* trials back was F (1,29) = 5.553, p = 0.025, η^2^p = 0.161). Consequently, we realised that since learning rates, unlike WSLS scores, take into account the history of outcomes, they are more suitable for analysing this data. To avoid duplicating analyses and creating an unwieldy, difficult-to-read, paper we focus here on the learning rate analysis only.

Following the Reviewer’s recommendation, we have added the analyses of ζ (social/non-social balance) values to the main text (lines 314-320, pg. 13).

“Linear mixed models, with fixed factors group and drug, and random intercepts for subject, were also used to explore drug effects on ζ values (representing the relative weighting of primary/secondary information) and β values. For ζ there were no significant main effects of drug (F (1, 29) = 1.941, p = 0.174, σx¯=-0.07 (0.050), t = -1.390, CI = [-0.170 – 0.003]) or group (F (1, 51) = 0.184, p = 0.669, σx¯=0.020 (0.040), t = 0.430, CI = [-0.070 – 0.100]), nor drug by group interaction (F (1, 29) = 0.039, p = 0.845, σx¯=-0.001 (0.050), t = -0.200, CI = [-0.110 – 0.090]). Similarly, our analysis of β values revealed no main/interaction effect(s) of drug, group, or drug by group (all p > 0.05).”

To illustrate how robust our results are we have added the following analysis, which follows a commonly used procedure (e.g., Cook et al., 2019; Browning et al., 2015), to Appendix 3 (p. 11):

“In addition, to formally test our model’s predictions of choice behaviour, for each participant we calculated the average value that the model estimated for the options chosen by the participant (collapsed across HAL and PLA conditions), and the average value that the model estimated for the options that were not chosen by the participant. If our model was accurately describing participants’ choice behaviour, then average estimated values for chosen options should be significantly higher than for the unchosen options. Indeed, a paired samples t-test illustrated that, model-derived values for chosen options (x¯(σx¯) = 0.607 (0.008)) were significantly greater than values for unchosen options (x¯(σx¯) = 0.393 (0.008); t(30) = 12.558, p < 0.001).”

To further illustrate how robust our results are we have clarified our simulations and added a second additional analysis to Appendix 3 (p. 11):

“To ensure that parameter estimates could be recovered, we simulated response data for each participant, based on estimated model parameters, using the function tapas_simModel.m from the TAPAS toolbox. Model parameters were subsequently estimated from simulated data and averaged over 100 iterations for each participant, separately for HAL and PLA conditions. All recovered parameters correlated significantly with estimated parameters under both HAL (α_primary:_ r = 0.991, p < 0.001; α_secondary:_ r = 0.961, p < 0.001) and PLA (α_primary:_ r = 0.975, p < 0.001 ; α_secondary:_ r = 0.984, p < 0.001) treatment conditions. A RM ANOVA on recovered parameters showed the same pattern of results as with estimated parameters including a significant interaction effect for our main interaction of interest (drug by information source: (F (1,29) = 4.027, p = 0.054, η_p_^2^ = 0.122)).”

4) While the manuscript is very tightly written, I would have appreciated a little more signposting for the secondary analyses. The main effect in Figure 3 seems clear enough, but there are so many tests and secondary analyses (e.g., ll. 267-292) that I found I got bogged down around the analyses leading to Figure 4. Even a few extra comments to reinforce the big picture would be appreciated.

We thank the Reviewer for highlighting this we have now made two important changes that we hope have improved the clarity of our manuscript. First, we have added the following paragraph describing our analysis strategy (pg. 8-9). We hope that this provides readers with a roadmap that they can use to navigate the Results section:

“We used the following strategy to analyse our data. First, we sought to validate our manipulation by testing (under PLA) whether participants in both the individual-primary and social-primary groups learned in a more optimal fashion from the primary, versus secondary, source of information. Next, we tested our primary hypothesis that both social and individual learning would be modulated by haloperidol when they are the *primary* source of learning, but not when they comprise the *secondary* source. To do so we estimated learning rates for primary and secondary sources of information, for each group (social-primary, individual-primary), under HAL and PLA, by fitting an adapted Rescorla-Wagner learning model to choice data. To ascertain that our model accurately described choices we used simulations and parameter recovery. We used random-effects Bayesian model selection to compare our model with alternative models. These analyses provided confidence that our model accurately described participants’ behaviour. After testing our primary hypothesis, we explored the relationship between parameters from our computational model and performance. To accomplish this, we first used an optimal learner model, with the same architecture and priors as our adapted Rescorla-Wagner model, to assess the extent to which haloperidol made participants’ learning rates more (or less) optimal. Finally, we regressed estimated model parameters against accuracy to gain insight into the extent to which variation in these parameters (and the effect of the drug thereupon) contributed to correct responses on the task.” (p. 8-9, lines 222-238)

Second, we have reviewed and clarified lines 267-292 (now lines 288-313) as follows:

“We hypothesised an interaction between drug and (primary versus secondary) information source such that haloperidol would affect learning from the primary information source only, regardless of its social/individual nature. To test this hypothesis, we employed a linear mixed effects model with fixed factors information source (primary, secondary), drug (HAL, PLA), environmental volatility (volatile, stable) and group (social-primary, individual-primary) and dependent variable α(square-root transformed to meet assumptions of normality). We controlled for inter-individual differences by including random intercepts for subject. Including pseudo-randomisation schedule as a factor in all analyses did not change the pattern of results. The mixed model revealed a drug by information interaction (F (1, 203) = 6.852, p = 0.009, σx¯= 0.026 (0.010), t = 2.62, CI = [0.010 – 0.050])” (lines 288-313).

5) Optimal learner analysis – I did not find the details about the analysis in the methods/appendices – maybe I missed them? Could you highlight them?

We thank the Reviewer for highlighting this and apologise for this omission. We have included details of the Optimal learner model in Methods.

6) Optimal learner analysis – this seems like a very interesting analysis step, why did you not use it in the HAL condition to characterize its effects in this respect?

We thank the Reviewer for this query. We primarily used the optimal learner analysis to show that the task manipulation (orthogonalizing social/individual and primary/secondary) had modulated participants’ behaviour in the expected direction. With respect to effects of haloperidol on performance in learning paradigms, studies have reported both improvements and decrements in task performance under haloperidol (Bolstad et al., 2015; Clos et al., 2018, 2019; Fallon et al., 2019; Frank and O’Reilly, 2006; Zirnheld et al., 2004). Therefore, we did not have strong a priori hypotheses regarding the effects of haloperidol on optimal learning rates and wanted to restrict the extent to which we explored the data.

7) Learning rate estimates minus optimal learning rates – while optimal learning rates are around 0.11, and estimated learning rates are around 0.4, the difference between them is reported to be around 0 (Figure 4). I am probably missing something here but I am not sure what.

We thank the Reviewer for highlighting this. Estimated learning rates have been square-root transformed (see line 674 – Methods and in the legend for Figure 3) to meet assumptions of normality for parametric testing. However, for calculating the ‘distance from optimal scores’, we used untransformed learning rates (see line 355). We have now highlighted this on lines 292-293 and have added summary statistics for the raw (untransformed) learning rates, to improve the clarity of the manuscript. (Table II, referred to on p. 25, line 674).

8) There are some discrepancies in the description of results of the optimal β learner results. For example, it indicates that the average β for primary is 0.872 (line 232), but the average of social-primary is 0.833 and the average of ind-primary is 0.477, whose average is ~0.6 – can you explain this?

We apologise for any lack of clarity here. In the part of the manuscript reporting results from the optimal β learner analysis**,** social-primary and individual-primary do not refer to the β values for primary learning only within the social-primary and individual-primary groups respectively; rather these values refer to the β values for social-primary and individual-primary groups averaged over both primary and secondary conditions. We have updated the manuscript (lines 250-255) to reflect this. In the process of doing so, we realised that the manuscript refers to optimal β values under placebo only, while the figure (Figure 2) reflected β values averaged across both conditions (HAL,PLA). The figure has been updated to correct this. We thank the Reviewers for their queries and hope our changes significantly improve the clarity of our manuscript.

Also, you indicate that there is no interaction effect, but the differences observed in the graph (Figure 2) and the reported effects suggest that there may be a significant effect (social primary-secondary > ind primary-secondary) ?

We thank the Reviewer for highlighting this. Our analysis here revealed a main effect of group (individual-primary vs social-primary) and a main effect of information source (primary vs secondary) with no interaction between group and information source. Bayesian analyses supported this lack of a group by information interaction (BF_excl_ = 2.844). Upon reflection we wonder whether the source of confusion is the significance stars that we added to Figure 2. We have now removed the lines and stars and, instead, have added a fuller description of the results to the figure legend itself. We hope that this change reduces any potential for confusion.

9) Learning rates effects – it is not clear whether volatility had any effect, and how it is incorporated in analysis – for example in Figure 3 and the text (line 280). You state α_primary, but not whether this is during volatile or stable periods. Did you average the learning rates between periods? Used random effect for these observations? This is also the case for the optimal learning rate analysis.

We thank the Reviewer for this important query. Based on previous work (Cook et al., 2019; Behrens et al., 2007), our a priori planned analysis strategy incorporated volatility (see Pre-registration- Analysis Plan). When we came to analyse the data, to ascertain that this was indeed a suitable strategy, we also carried out a model comparison (see Appendix 3, p. 6-7) in which we compared the fit of models which did (Models 3, 4, 7 and 8) and did not (Models 1,2, 5 and 6) model effects of the volatility manipulation. Model 4 was the winning model. Thus, we used Model 4 – which estimates separate learning rates for learning in volatile and stable environments – in all ensuing analyses. Interestingly, although for a good fit to the data it is important to model volatility in the context of the current paradigm, we did not find any main effects of, or interactions involving, volatility in our primary analyses (e.g., line 298). Consequently, to unpack our significant main effects and interactions we collapsed across volatile and stable learning rates, meaning that α_primary_ refers to learning rate from the primary source of information, averaged across volatile and stable phases – i.e., all trials – and α_secondary_ refers to learning rate from the secondary source of information, averaged across volatile and stable phases (as previously, referring to all trials).

This was also the case for the optimal learning rate analysis: αdiff_primary refers to the difference between learning rate from the primary source of information and optimal learning rate, averaged across volatile and stable phases and αdiff_secondary refers to the difference between learning rate from the secondary source of information and optimal learning rate, averaged across volatile and stable phases. The optimal learning rate analysis was carried out using a standard RM ANOVA. To bring these results in line with our main analysis, we have now corrected this and report a linear mixed effects analysis for optimal difference scores (αdiff), including subject as a random effect. Thus, to respond to the Reviewer’s query about random effects, all learning rates analyses used a linear mixed effects model with subject included as a random effect.

“A linear mixed model analysis on αdiff values with factors group, drug, volatility and information source, and random intercepts for subject was conducted. A significant interaction between drug and information source was observed (F (1, 203) = 4.895, p = 0.028, σx¯= 0.019 (0.010), t = 2.212, CI = [0.000 – 0.040]) (Figure 4). Planned contrasts showed that, for primary information, αdiff_primary  (collapsed over volatile and stable) was higher under PLA (x¯(σx¯) = 0.052 (0.023)) compared with HAL (x¯(σx¯) = 0.009 (0.028); z(30) = -1.806, p = 0.071). In contrast, αdiff_secondary  was lower under PLA (x¯(σx¯) = -0.011 (0.023)) compared with HAL (x¯(σx¯) = 0.021 (0.021)); z(30) = 1.323, p = 0.186. Learning rates for learning from the primary source were higher than optimal under placebo, with αdiff_primary  significantly differing from 0 (one-sample t test; t(30) = 2.259, p = 0.0310). Haloperidol reduced learning rates that corresponded to learning from the primary source, thus bringing them within the optimal range, with αdiff_primary  not significantly differing from 0 under haloperidol (one-sample t test; t(30) = 0.319, p = 0.752). Consequently, under haloperidol relative to placebo, learning rates were *more optimal* when learning from primary sources”. (Lines 355-368)

Finally, we have updated the legend for Figures 3 and 4 to clarify that α values refer to both volatile and stable phases.

“Figure 3. Learning rate (α) estimates for learning from primary and secondary information across all trials (averaged across volatile and stable phases).”

“Figure 4. Learning rate estimates minus optimal learning rates. There was a significant interaction between information and drug, with α_primary_ scores significantly higher than optimal estimates under placebo but not under haloperidol. Data points indicate α− α_optimal_ values for individual participants (n = 31) across all trials (averaged across volatile and stable phases).”*Additional Comments*

1) I found the lack of a significant main effect of drug on accuracy very surprising. Other behavioral differences such as response times between the social-primary and individual-primary groups due to haloperidol relative to placebo could have potentially affected choices in different ways. Were there any differences in accuracy with respect to response times due to haloperidol vs. placebo?

We thank the Reviewer for inviting us to further reflect on this. A RM ANOVA, with drug and group as fixed factors, and reaction time as the DV, revealed a marginally significant main effect of haloperidol on reaction times (F (1,29) = 3.810, p = 0.061, η^2^p = 0.116), with slower reaction time (RT) under haloperidol (x¯(σx¯) = 1.580 (0.147)) versus placebo (x¯(σx¯) = 1.242 (0.150)). There was no main effect of group, or group by drug interaction.

To ensure that there were no differences in accuracy with respect to response times, whereby variable measures could lead to contradictory conclusions about the effect of group, we have performed an analysis which could account for possible speed-accuracy trade-offs. Inverse efficiency scores (IES; Townsend and Ashby, 1978,1983), were used, which combine speed and accuracy into a single score. IES comprises the RT divided by the proportion of correct responses. IES were calculated for each treatment condition and compared between groups. A RM ANOVA, with drug and group as fixed factors, and IES scores as the DV, revealed that the main effect of haloperidol on IES approached significance (F (1,29) = 4.810, p = 0.072, η^2^p = 0.108) with lower RTs under haloperidol (x¯(σx¯) = 2.708 (0.286)) versus placebo (x¯(σx¯) = 2.084 (0.270)), when corrected for percentage accuracy. As with RT scores, there was no main effect of group, or group by drug interaction. Indeed, Bayesian analysis provided evidence against a main effect of group (BF_excl_ = 1.499) or a drug by group interaction (BF_excl_ = 1.188). Thus, whilst the effects of drug on RT are marginally significant, we cannot provide any evidence to support the conclusion that differences in accuracy with respect to RT occur between the social-primary and individual-primary groups as a function of drug.

2) There were social-primary group and individual-primary group. I am curious if there were fluctuations (within the group) on how much people relied upon social versus individual sources of information across different trials, and if these fluctuations could have differently influenced the effects of haloperidol.

We thank the Reviewer for inviting us to reflect on this. We were unable to detect trial-wise fluctuations in the relative weighting of social and individual information in our paradigm. The social weighting parameter, ζ represents the relative weighting of primary and secondary sources of information, with higher values indicating a bias towards the over-weighting of secondary relative to primary. However, this parameter does not get updated in a trial wise manner but is rather a single estimated value which varies between subjects, not across different trials.

Although we could not investigate this on a trial-by-trial basis, to follow through the Reviewer’s line of thinking we examined individual differences in ζ values and whether ζ interacted with our main effects of interest. We split participants into two groups based on the median ζ and included ζ (high/low) as a predictor in a linear mixed effects model. Importantly, the significant drug by information interaction effect remained significant with the inclusion of ζ group (F(1,183) = 4.933, p = 0.028) and no significant main/interaction effects of ζ were observed (all p > 0.05). Thus, the effect of haloperidol on learning rates did not vary as a function of individual differences in the extent to which participants relied upon social versus individual sources of information.

3) Using the TAPAS toolbox, which has explicit volatility estimation model similar to the one used by Behrnes et al., 2008, raises the question whether using such model which directly evaluates volatility may be more useful in analysis of learning pattern here. Put in other words, would you lose anything if volatility was not changed during the task – just a stable period of 0.7 probability of accurate advice/blue, for example?

The decision to use a task which features both stable and volatile periods was made for two reasons. First, having stable and volatile periods that happen at different times for the primary and secondary sources of information (as in Behrens et al., 2008) ensures that we can estimate learning rates for the two information sources (primary and secondary) simultaneously, and also in a dissociable manner. Second, we had a priori hypotheses involving adjustment of learning to volatility as a function of dopamine manipulation (see Pre-registration – Hypothesis). That is, in previous work we found that methylphenidate, a dopamine reuptake inhibitor, affected participants’ ability to adjust learning rate in response to changes in environmental volatility and that this effect was restricted to learning from the primary (in this case “individual”) but not secondary (in this case “social”) source of information (Cook et al., 2019, *eLife*). We thus aimed to use the same design as Cook et al., 2019, plus an additional manipulation that enabled us to orthogonalize the primary/secondary and social/individual nature of the information source. As noted in our response to comment 9 (above), when analysing the data, we compared various models (including some models which ignored the volatility manipulation) and found that Model 4 which accounts for the stable and volatile phases provided the best fit to the data. Thus, our decision to include volatility in the design and analysis was based on both our a priori hypotheses and (post data collection) model comparisons.

The Reviewers are possibly wondering why we did not use a Hierarchical Gaussian Filter model, as has been employed in some related literature (e.g., Diaconescu et al., 2014, 2017). However, we found that with the current dataset this model frequently failed to converge. Furthermore, since our task features a novel manipulation (enabling orthogonalization of primary/secondary and social/individual), we were unclear as to what the most appropriate priors for the parameters should be and therefore, concerned that the lack of a priori priors would create too much flexibility in our analysis if we were to pursue this route.

Since we did not observe main/interaction effects involving volatility, subsequent (partial) replications could abandon the volatility manipulation as long as they ensured that the probability schedules underpinning the primary and secondary learning sources are different enough that one can dissociate learning rates for the two information sources from each other.

4) I find the result that primary learning from social info is better (more similar to optimal learning – β value analysis and Figure 2) than primary learning from non-social information very interesting. It indicates that, when cast as primary task, people are very attuned to social information and can make a good use of it. This is something we now see in works where people learn about people's traits, such as honesty ( Bellucci 2019). I wonder if this result and the prospect of recasting social tasks can be given some more attention.

We thank the Reviewer for highlighting this and agree that this result could be given greater attention. We have added the following to the Discussion (p. 17, lines 427-438):

“The first part of our analysis illustrated that our manipulation produced the expected effect: when social information was first in the temporal order of events, highly salient and directly related to reward feedback participants learned in a more optimal fashion from this source of information. Such a result may be a surprise to some since one might think that, relative to learning from one’s own experience, learning from others will always take a “backseat”. Here we clearly demonstrate that, when cast as the primary task, participants can make good use of social information. This paradigm may comprise a step towards developing a system to support accelerated social learning. Future studies could, for instance, investigate whether similar manipulations can be used to improve learning *about* (as opposed to *from*) other individuals. Since temporal order, saliency and reward feedback were manipulated simultaneously we cannot determine which manipulation is the most influential. Future work may therefore also seek to manipulate these factors independently to establish the most effective method for promoting social learning."